# Summary of Changes
# Between this New Submission and the Previous Submission

We would like to thank the reviewers for the insightful comments and constructive suggestions. We are delighted that **ALL** reviewers recognize the **practical significance** of our *cross-ISA binary code translation* work, which makes an important contribution to detecting malware across ISAs by leveraging a model trained on a high-resource ISA (X86-64), effectively addressing the data scarcity challenge of low-resource ISAs. We are also encouraged by the reviewers' acknowledgment of the **novelty** of this work and its ability to fill gaps in the current state of the art (Reviewers DKbR and FiFn), its coverage of **multiple ISAs** (Reviewers DKbR, FiFn, and yy4t), the value of **unsupervised learning** (Reviewer yy4t), the **improved performance** (Reviewers FiFn and yy4t), and the **clear interpretation and explanation** of how and why our model evolves (Reviewer C5ox). Additionally, we appreciate the positive feedback on the quality of writing (Reviewer FiFn).

We first summarize the **newly added experiments** as follows.

- Expanded the malware datasets and conducted all experiments using these larger datasets (Reviewer DKbR).
- Utilized CNNs for malware detection and presented the detection results (Reviewer DKbR and FiFn).
- Included the MIPS ISA in the evaluation (Reviewer DKbR).
- Trained the baseline model, UnsuperBinTrans, on the additional ISA, and compared its performance with our model (Reviewer FiFn).

In addition to the new experiments, this submission has been carefully revised to address all reviewers' comments and concerns. Detailed responses and changes for each reviewer comment are provided below.

===================== **Response to Reviewer DKbR** =====================

All the changes made in response to Reviewer DKbR's comments are highlighted in Blue.

**1. Reviewer's Comment**:
"*Small malware datasets*."

**Authors' Response**:
We have expanded the malware datasets and conducted experiments using these larger datasets. Specifically, we collected 2140,1740,1581,58, 1430 and 2 malware samples for the X86-64, i386, ARM32, ARM64, MIPS32, and s390X architectures, respectively. All experimental results have been updated accordingly, as detailed in **the Evaluation Section**.

-------------------------------------------------------------------------------------------------------------------

**2. Reviewer's Comment**:

*"Unclear description. MALTRANS is introduced as an ISA-to-ISA translation tool and presented as such in Figure 1, but later, it is also described as capable of detection. Clarification is needed to delineate its functions." "Figure 2 shows the MALTRANS architecture; however, the flow is not entirely clear." "In the paragraph "Comparison with Optimal Model," the dataset usage is unclear."*

**Authors' Response**:
MALTRANS is designed for ISA-to-ISA translation. The malware detection capability is provided by the LSTM model, which analyzes assembly code translated by MALTRANS to detect malware. We have revised **the caption of Table 3** (**Line 378**) to clarify this point.

We have replaced the original **Figure 2** with two separate subfigures (**Line 216** to **Line 223**) that illustrate how denoising auto-encoding and back translation are employed to train MALTRANS. The presentation in **Section 4.4** has also been revised accordingly to align with these changes.

In the "Comparison with Optimal Model" paragraph, for x86-64, i386, and ARM32, we used an 80/20 split for training and testing. For ARM64 and s390x (this is not as popular as x86-64 and ARM32), however, we could only obtain 58 and 2 malware samples, respectively, from VirusShare.com (after deduplication). We used 46 ARM64 and 1 s390x samples for training and 12 ARM64 and 1 s390x samples for testing. While these sample sizes are limited, they reflect the value of MALTRANS: by translating code from a low-resource ISA to a high-resource ISA, we can leverage models trained on the high-resource ISA, addressing data scarcity challenges that would otherwise hinder robust detection for low-resource ISAs. We have revised the **"Task-Specific Training Dataset" and "Task-Specific Testing Dataset" parts in Section 5.4** (**Line 401** to **Line 422**) to clarify this.

---

**3. Reviewer's Comment**:
*"Experiments performed on a single deep learning model."*

**Authors' Response**:
We have utilized CNNs for malware detection and included the detection results in **Appendix E** (**Line 803**).

---

**4. Reviewer's Comment**:
*"Absence of the MIPS ISA."*

**Authors' Response**:
We have collected 1430 MIPS malware samples and included them in the evaluation. The revised results include updates to the vocabulary size (**Table 1; Line 424**), BLEU scores (**Table 2; Line 332**), malware detection results (**Table 3; Line 378**), and the hyperparameter study (**Tables 4 and 5; Line 486** and **Line 494**). The presentation has been revised accordingly to reflect these updates.

---

**5. Reviewer's Comment**:
*"The acronyms for BLEU, AUC, and F1-score are not expanded. Although they are widely recognized, it would be helpful to describe each acronym and provide brief descriptions."*

**Authors' Response**:
We have included the descriptions of BLEU score in **Line 355** and AUC/F1-score in **Line 420**.

======================= **Response to Reviewer FiFn** =======================

All the changes made in response to Reviewer FiFn's comments are highlighted in Magenta.

**Reviewer's New Comment after Rebuttal**:
"*However, my concerns about limited novelty compared to UnsuperBinTrans remain, and UnsuperBinTrans already achieved some of the claimed contributions. Therefore, I will maintain my original scores.*"

**Authors' Response**:
Thank you for your kind response, and we apologize for not clarifying this earlier. Below, we summarize the key novelties of MalTrans:

1. **Enhanced Normalization Rules:** MalTrans introduces new normalization rules for assembly code, which differ significantly from those used by UnsuperBinTrans. Specifically, normalization rules R1 and R2, which address issues with dummy names generated by IDA Pro and normalize function names, are not applied by UnsuperBinTrans. The absence of these rules results in numerous out-of-vocabulary words during testing, potentially degrading translation performance. Table 4 illustrates the significant impact of R1 and R2 on malware detection performance.
2. **Application to Malware Detection:** UnsuperBinTrans has not been previously applied to the malware detection task, leaving its effectiveness in this domain uncertain. As shown in Table 3(a), when UnsuperBinTrans is applied to malware detection, it performs poorly. This could be due to its unsuitable normalization rules and less effective model architecture. In contrast, MalTrans achieves superior malware detection across ISAs, thanks to its improved model architecture and tailored normalization schemes.
3. **Broader ISA Coverage:** While UnsuperBinTrans is limited to two ISAs (x86-64 and ARM 32), MalTrans extends the evaluation to a wider range of ISAs, demonstrating both broader applicability and improved performance.

We hope this clarifies the distinct advantages of MalTrans compared to UnsuperBinTrans.

-------------------------------------------------------------------------------------------------------------

**1. Reviewer's Comment**:
"*Among the contributions claimed in the paper, similar contributions were made in UnsuperBinTrans. The novelty is the model architecture, normalization scheme, and availability for more ISAs.*"

**Authors' Response**:
We have revised **the Introduction Section** (the second bullet in the Contribution list; **Line 82**) to highlight that our primary contribution is in code translation to support malware detection, effectively addressing the data scarcity challenge that would otherwise impede robust detection for low-resource

ISAs. Additionally, we have revised **the Conclusion Section** in **Line 537** to compare our work and UnsuperBinTrans.

---

**2. Reviewer's Comment**:
"*Assessment for s390x does not seem to be fair since there are only 2 malware samples*."

**Authors' Response**:
We have revised the **"Result Analysis" part in Section 5.4** (**Line 427**) to clarify that the s390x results primarily demonstrate MALTRANS's adaptability in low-resource conditions, and future work will focus on including a larger s390x dataset for a more comprehensive evaluation.

---

**3. Reviewer's Comment**:
"*It seems that the testing data sets are not consistent between Table 3 (a) and (b) for ARM64 and s390x. So the accuracies obtained cannot be directly compared*."

**Authors' Response**:
We have reconducted the experiments to ensure that only 20% of malware samples across the five ISAs are translated by MALTRANS and tested using the x86-64-trained LSTM model for consistency with Table 3(b). We have revised the **"Task-Specific Training Datasets" and "Task-Specific Testing Datasets" in Section 5.4** (**Line 401**) to detail how these datasets are constructed. The **"Result Analysis" part in Section 5.4** (**Line 424**) and **Table 3(a)** (**Line 380**) have been updated accordingly. Additionally, the results where the LSTM model is trained on X86-64 and tested on *all* malware samples in the other five ISAs have been moved to **Appendix F** (**Line 871**).

---

**4. Reviewer's Comment**:
"*Do we actually need to go through a complex code translation process when we can get good results with 265 data points?*."

**Authors' Response**:
We have clarified this in the **"Comparison with Optimal Model" part in Section 5.5** (**Line 478**), explaining that, in an extreme case where only one binary is available for a given ISA, it is still possible to detect whether it is malware using the x86-64-trained model, highlighting the necessity of translation to address the data scarcity issue.

---

**5. Reviewer's Comment**:
"*Can we get better BLEU scores if we have more data samples? Are these BLEU scores for 1-gram?*"

**Authors' Response**:
We have evaluated the adequacy of our mono-architecture datasets, with details in **Appendix B** (**Line 780**). We have clarified in **the first paragraph of Section 5.3** (**Line 358**) that the BLEU scores are computed as an average of unigram, bigram, trigram, and 4-gram precision.

------------------------------------------------------------------------------------------------------

**6. Reviewer's Comment**:
"*It would be better if it is possible to provide numbers for UnsuperBinTrans for other ISAs although it only focused on ARM32→X86. It should be trivial and a good comparison to do. Any possibility?*"

**Authors' Response**:
We have trained UnsuperBinTrans on the same mono-architecture training datasets for the additional ISA pairs, as described in **the fourth paragraph of Section 5.3** (**Line 372**). The results, including BLEU scores (**Table 2**; **Line 332**) and malware detection results (**Table 3**; **Line 378**), have been reported accordingly.

------------------------------------------------------------------------------------------------------

**7. Reviewer's Comment**:
"*Why the accuracy results are better for ARM32→X86 although the BELU score is higher for ARM64→X86? Any explanation for that?*"

**Authors' Response**:
The BLEU score measures n-gram overlap between translated and reference code, indicating translation quality based on structural similarity. However, downstream malware detection depends more on preserving code semantics rather than exact n-gram matches. This may explain why ARM32→X86 achieves higher detection accuracy despite a lower BLEU score, as the translated ARM32 code may better retain semantic features relevant to detection.

------------------------------------------------------------------------------------------------------

**8. Reviewer's Comment**:
"*It would be ideal to compare the performance of malware detection using classical machine learning techniques as a comparison with LSTM. Can that be done?*"

**Authors' Response**:
We have utilized CNNs for malware detection and included the detection results in **Appendix E** (**Line 804**).

------------------------------------------------------------------------------------------------------

**9. Reviewer's Comment**:
"*What would be the impact on the accuracy if few-shot learning is done? What should be the ideal ratio of data for 2 ISAs?*"

**Authors' Response**:
Few-shot learning could potentially improve accuracy. However, since our focus is on unsupervised learning, we did not use labeled code samples for training MALTRANS. We leave it as future work.

===================== **Response to Reviewer yy4t** =====================

All the changes made in response to Reviewer yy4t's comments are highlighted in Red.

**1. Reviewer's Comment**:
"*How does this work compare and contrast with the related works mentioned ([7], [9], [11], [12])?*"

**Authors' Response**:
We have discussed this in **Appendix H** (**Line 1006**).

-------------------------------------------------------------------------------------------------------------------

**2. Reviewer's Comment**:
"*How can the paper justify the statement that UnsuperBinTrans is the first and only existing work without comparing or contrasting with other relevant works mentioned in the relevant references?*"

**Authors' Response**:
We have revised **Section 2.3** (**Line 147**) that UnsuperBinTrans is the first and only existing work focused on binary code translation by leveraging deep learning techniques.

-------------------------------------------------------------------------------------------------------------------

**3. Reviewer's Comment**:
"*How practical are the two principles utilized for BPE merge times?*"

**Authors' Response**:
The two principles are derived from both empirical experiments and theoretical insights discussed in prior research. We have revised the **"Byte Pair Encoding (BPE)" part in Section 5.2** (**Line 322** and **Line 341**)to clarify this.

-------------------------------------------------------------------------------------------------------------------

**4. Reviewer's Comment**:
"*In reality, would a vocabulary size discrepancy exist unless the reviewer is missing something?*"

**Authors' Response**:
Yes, a vocabulary size discrepancy (> 15%) may exist. If the vocabulary size of one ISA is significantly larger (or smaller) than that of another, a subset of the vocabulary from this ISA (or the other) may be selected for training MALTRANS to address the imbalance.

-------------------------------------------------------------------------------------------------------------------

**5. Reviewer's Comment**:
"*Is there supporting evidence to back the recommendation that "The vocabulary size of each ISA is recommended to be <12K"?*"

**Authors' Response**:

We have revised **the second bullet point in the "Byte Pair Encoding (BPE)" section of Section 5.2** (**Line 341**) to clarify this. The 12K threshold was determined empirically as the optimal balance between model performance and computational efficiency.

---------------------------------------------------------------------------------------------------------------------

**6. Reviewer's Comment**:
*"What are the final evaluation losses for the training of each ISA?"*

**Authors' Response**:
For each ISA pair, we trained the translation model until the evaluation loss dropped below 0.5, as presented in **the "Training Details" part of Section 5.2** (**Line 350**).

---------------------------------------------------------------------------------------------------------------------

**7. Reviewer's Comment**:
*"How many parameters does the model have if the training takes around one day?"*

**Authors' Response**:
We have presented the parameters of the shared encoder, source decoder, target decoder in **the "Training Details" part of Section 5.2** (**Line 350**).

---------------------------------------------------------------------------------------------------------------------

**8. Reviewer's Comment**:
*"Why did the paper utilize 4-gram precision? Why not higher than 4-gram precision?"*

**Authors' Response**:
We have clarified in **the first paragraph in Section 5.3** (**Line 355**) that the BLEU scores are computed as an average of unigram, bigram, trigram, and 4-gram precision.

We did not consider n-grams higher than 4-grams for the following reasons. (1) Higher n-grams (5+ grams) become increasingly sparse. This issue is exacerbated in code translation, where instruction sequences are typically shorter than natural language sentences, making higher n-grams less reliable as evaluation metrics. (2) The original BLEU metric proposed by [16] demonstrated that 4-gram BLEU provides an optimal balance between accuracy and computational efficiency. Beyond 4-grams, the benefits diminish due to increased computational costs and reduced reliability.

---------------------------------------------------------------------------------------------------------------------

**9. Reviewer's Comment**:
*"How does the translation quality compare with other works such as [11] and [12]?"*

**Authors' Response**:
The work in [11] utilizes neural machine translation techniques for binary code similarity comparison but does not perform binary code translation across ISAs. Specifically, it employs only the encoder of a neural machine translation model to generate embeddings for two pieces of binary code, and measures similarity based on embedding distance. In contrast, our work focuses on translating binary code across different ISAs.

The work in [12] focuses on source code translation (e.g., C to Java), which differs significantly from binary code translation (e.g., assembly code from x86-64 to ARM32).

As the objectives and languages differ, a direct comparison of translation quality is not applicable.

---------------------------------------------------------------------------------------------------------------

**10. Reviewer's Comment**:
"*Should the reported results include an error range?*"

**Authors' Response**:
Apologies for the delay. Due to time constraints, we have not completed this experiment. We will include the error range in the final revision.

---------------------------------------------------------------------------------------------------------------

**11. Reviewer's Comment**:
"*discuss the scalability issue of the proposed system and how the system can be adapted to handle the packed or obfuscated malware.*"

**Authors' Response**:
We have presented the translation time in **the last paragraph of Section 5.3** (**Line 377**) and discussed how to handle packed or obfuscated malware in **Appendix H (Line 1020)**.

---------------------------------------------------------------------------------------------------------------

**12. Reviewer's Comment**:
"*more details about the model architecture, data preprocessing, and hyperparameters*"

**Authors' Response**:
We have detailed the data preprocessing in **Appendix A (Line 756)**. We have detailed model architecture and hyperparameters in **Appendix C** (MALTRANS; **Line 810**), **Appendix D** (LSTM; **Line 826**) and **Appendix E** (CNN; **Line 837**).

======================= **Response to Reviewer C5ox** =======================

All the changes made in response to Reviewer FiFn's comments are highlighted in Violet.

**1. Reviewer's Comment**:
"*Motivation is weak… Is it true that IoT malware are so numerous to outmatch all other domains (like Windows and Android)?*"

**Authors' Response**:
We have revised the **"Motivation" part in Section 4** (**Line** 153). The motivation for this work stems from the data scarcity of malware, which is a significant challenge for IoT. Due to the heterogeneity of IoT devices, a variety of ISAs are used in their development. However, this diversity has led to a lack

of sufficient malware data for many ISAs, hindering effective malware detection. In contrast, Windows and Android, which primarily operate on x86 and ARM architectures, do not face the issue of data scarcity.

Furthermore, malware detection for platforms like Windows and Android is a well-studied problem, with numerous mature solutions available. For IoT, however, malware detection remains underexplored, and critical issues, such as the scarcity of malware data for certain ISAs, remain unresolved.

Notably, IoT malware incidents surged by 87% in 2022 compared to 2021, reaching 112.3 million cases (link). This sharp rise highlights the increasing security threats facing IoT ecosystems.

-------------------------------------------------------------------------------------------------------------

**2. Reviewer's Comment**:
"*How the authors computed the BLEU score for their experiments?*"

**Authors' Response**:
We have revised **the first paragraph in the "Translation Results" section of Section 5.3** (**Line 356**) to explain how we compute the BLEU scores.

-------------------------------------------------------------------------------------------------------------

**3. Reviewer's Comment**:
"*Which specific ISAs are the authors targeting with their experiments?*"

**Authors' Response**:
We apologize for the confusion. The term "x86" used in the paper actually refers to x86-64. We have carefully revised the paper to eliminate this ambiguity.

-------------------------------------------------------------------------------------------------------------

**4. Reviewer's Comment**:
"*Missing examples. Can the authors submit examples of their translations? This might help understand better the quality of the translation.*"

**Authors' Response**:
We have included some examples of our translations in **Table 12 and 13 in Appendix F** (**Line 931** and **Line 972**).

# MALTRANS: UNSUPERVISED BINARY CODE TRANSLATION FOR MALWARE DETECTION

**Anonymous authors**

## ABSTRACT

Applying deep learning to malware detection has drawn great attention due to its notable performance. With the increasing prevalence of cyberattacks targeting IoT devices, there is a parallel rise in the development of malware across various Instruction Set Architectures (ISAs). It is thus important to extend malware detection capacity to multiple ISAs. However, training a deep learning-based malware detection model usually requires a large number of labeled malware samples. The process of collecting and labeling sufficient malware samples to build datasets for each ISA is labor-intensive and time-consuming. To reduce the burden of data collection, we propose to leverage the ideas and techniques in Neural Machine Translation (NMT) for malware detection. Specifically, when dealing with malware in a certain ISA, we translate it to an ISA with sufficient malware samples (such as X86-64). This allows us to apply a model trained on one ISA to analyze malware from another ISA. Our approach reduces the data collection effort by enabling malware detection across multiple ISAs using a model trained on a single ISA. We have implemented and evaluated the model on six ISAs, including X86-64, i386, ARM64, ARM32, MIPS32 and s390x. The results demonstrate its high translation capability, thereby enabling superior malware detection across ISAs.

## 1 INTRODUCTION

The impacts of malicious software are worsening day by day. Malicious software, or malware, refers to programs designed to harm, interrupt, or damage computers, networks and related resources Preda et al. (2008). Nowadays, numerous malware detectors have been developed Xie et al. (2024b;a), and their effectiveness largely depends on the techniques employed. Signature-based malware detection searches for the patterns belonging to known malware families Sathyanarayan et al. (2008), but it is often inaccurate in detecting modified or new malware. Behavioral analysis-based detection examines the execution behavior of programs to detect suspicious actions Liu et al. (2011), but it is unscalable.

Applying deep learning to malware detection has drawn great attention due to its notable performance. Existing deep learning-based approaches leverage neural networks, such as CNNs, RNNs, and LSTM, to identify malware Sewak et al. (2018); Gopinath & Sethuraman (2023). The high accuracy and adaptability of deep learning models make them particularly effective at detecting even sophisticated and previously unseen malware variants He et al. (2023); Lei et al. (2022); Zhang et al. (2024).

**Challenge.** With the growing prevalence of cyberattacks targeting IoT devices, there has been a corresponding increase in the development of malware across various Instruction Set Architectures (ISAs). By creating malware capable of targeting multiple ISAs, attackers can maximize their reach and impact, enabling widespread attacks such as botnets that compromise numerous devices Davanian & Faloutsos (2022); Caviglione et al. (2020). Thus, it is crucial to extend malware detection capabilities to multiple ISAs. However, existing deep learning-based methods typically require a large number of malware samples for training. The process of collecting and labeling sufficient malware samples to build datasets for each ISA is labor-intensive and time-consuming.

**Our Approach.** Malware is typically a closed-source program, where the source code is usually unavailable. What we can access is the binary representation of malware. A binary, after being disassembled, is expressed in an assembly language. Given this insight, we propose to apply the ideas and techniques in Neural Machine Translation (NMT), which focuses on translating texts across human languages Artetxe et al. (2018) to reduce the burden of data collection.

When handling a binary in a given ISA (referred to as the *source* ISA), we translate it to an ISA with rich malware samples, such as X86-64, which we refer to as the *target* ISA. Once translated, we use a model trained on the *target* ISA to test the translated code. This approach facilitates malware detection across multiple ISAs using a model trained solely on the target ISA, eliminating the need for extensive malware samples in other ISAs.

We design an unsupervised binary code translation model called MALTRANS, which can translate binaries across ISAs. MALTRANS contains a shared encoder for both ISAs and a separate decoder for each ISA. It operates in a completely unsupervised manner, eliminating the need for parallel datasets. Importantly, the training of MALTRANS does not require any malware samples and relies only on binaries compiled from the abundance of open-source programs. Despite never encountering any malware samples during training, MALTRANS is still capable of translating malware across ISAs with high translation quality.

**Results.** We have implemented our model MALTRANS, and evaluated its performance on six ISAs: X86-64, i386, ARM32, ARM64, MIPS32, and s390x. Our experiments show that MALTRANS achieves up to $0.4$ BLEU score for i386→X86-64, $0.32$ BLEU score for ARM32→X86-64, $0.34$ BLEU score for ARM64→X86-64, $0.35$ BLEU score for MIPS32→X86-64, and $0.37$ BLEU score for s390x→X86-64, while the baseline method UnsuperBinTrans Ahmad & Luo (2023) reaches much lower BLEU scores for these ISA pairs. Furthermore, we apply MALTRANS to the malware detection task, and the results are extremely encouraging: when a malware detection model is trained on X86-64 and transferred to detect malware in the other five ISAs, it achieves AUC scores of $0.996$, $0.981$, $0.915$, $0.920$, and $0.923$ for i386, ARM32, ARM64, MIPS32, and s390x, respectively. These results show that MALTRANS has superior translation quality, thereby enabling exceptional malware detection in multiple ISAs by translating binaries across ISAs. Below we highlight our contributions:

- We propose MALTRANS, a novel unsupervised approach to translate binaries across different ISAs. The training of MALTRANS does not require any malware samples, yet it is still capable of translating malware across ISAs and achieves high translation quality.

- By translating binaries from low-resource ISAs to a high-resource ISA, MALTRANS enables the detection of malware in low-resource ISAs using a model trained on the high-resource ISA, effectively addressing the data scarcity challenge that would otherwise hinder robust detection for low-resource ISAs.

- We have implemented the model and evaluated its performances on six ISAs: X86-64, i386, ARM32, ARM64, MIPS32, and s390x. The results demonstrate the model's high translation quality, enabling superior malware detection across ISAs. We plan to make the source code, trained model, and datasets publicly available.

## 2 RELATED WORK

### 2.1 MALWARE DETECTION

**Signature-Based Detection.** Traditionally, malware detection relied heavily on signature-based methods, where known patterns of malicious code are identified and stored in databases Sebastio et al. (2020). Tools like antivirus software use the signatures to scan files and detect malware Rohith & Kaur (2021); Al-Asli & Ghaleb (2019); Sathyanarayan et al. (2008); Behal et al. (2010). While effective against known threats, such methods struggle with new or polymorphic malware, which can change its code to evade detection.

**Behavior-Based Detection.** Behavior-based detection identifies malware by analyzing the behavior of programs at runtime Liu et al. (2011); Burguera et al. (2011); Aslan et al. (2021); Saracino et al. (2016). It looks for suspicious activities, such as unusual system calls or network behavior Burguera et al. (2011); Aslan et al. (2021). However, these approaches are unscalable and suffer from false-negative rates if the malicious behavior is not triggered during monitoring.

**Machine/Deep Learning-Based Detection.** In recent years, machine/deep learning has emerged as a powerful tool for malware detection. Machine/deep learning techniques can analyze vast amounts of data and learn to identify patterns associated with malware. Techniques such as decision trees Utku et al. (2018), Support Vector Machines (SVMs) Hasan & Rahman (2017), and neural networks Jeon et al. (2020), including models like CNNs, RNNs, and LSTM, have been widely applied in malware detection tasks Sewak et al. (2018); Gopinath & Sethuraman (2023); Wang et al. (2023a; 2024).

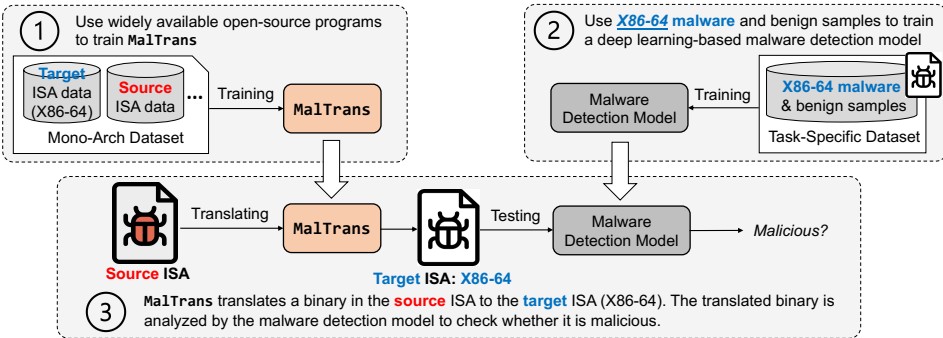

Figure 1: Applying MALTRANS to detect malware in a source ISA using a malware detection model trained on the target ISA (X86-64).

## 2.2 SOURCE CODE TRANSLATION

Source code translation refers to the process of converting code written in one programming language into another. Early work in source code translation uses *transpilers* or *transcompilers* Andrés & Pérez (2017); Tripuramallu et al. (2024), which rely on handcrafted rules. However, they produce unidiomatic translations that prove hard for human programmers to read. Another issue is incomplete feature support. For instance, a feature available in the source language might not have a direct equivalent in the target language, leading to functional gaps or the need for workarounds.

Recent advancements in deep learning have introduced new approaches to source code translation Roziere et al. (2020); Weisz et al. (2021); Lachaux et al. (2020). But as malware is closed-source where the source code is usually unavailable, source code translation does not work for malware.

## 2.3 PROGRAM ANALYSIS-BASED BINARY CODE TRANSLATION

Several approaches leverage program analysis techniques to convert binary code from one ISA to another. They can be broadly categorized into *static analysis-based translation* and *dynamic analysis-based translation*. Static analysis-based translation analyzes and translates the binary code before execution Shen et al. (2012); Cifuentes & Van Emmerik (2000). However, ensuring that all paths are accurately translated can be challenging. Dynamic analysis-based translation performs translation during execution Chernoff et al. (1998); Ebcioglu et al. (2001), but it introduces runtime overhead and requires sophisticated runtime environments, which can increase complexity and resource consumption. Additionally, both approaches face challenges related to *architecture-specific features* and encounter difficulties in achieving accuracy, performance, and compatibility.

In this work, we propose unsupervised binary code translation inspired by recent advances in neural machine translation. This represents a new and emerging direction. To the best of our knowledge, UnsuperBinTrans Ahmad & Luo (2023) is the first and only existing work that applies deep learning techniques to binary code translation. Our evaluation demonstrates that our model outperforms UnsuperBinTrans in binary code translation and achieves superior malware detection across ISAs.

## 3 MOTIVATION AND OVERVIEW

**Motivation.** IoT malware incidents surged by 87% in 2022 compared to 2021, reaching 112.3 million cases Sonicwall (2023). The heterogeneity of IoT devices introduces a wide variety of ISAs used in their development, driving a parallel increase in malware targeting multiple ISAs. This highlights the critical need to extend malware detection capabilities across ISAs. However, existing deep learning-based methods typically rely on large datasets of malware samples for training. The diversity of ISAs exacerbates this challenge, as sufficient malware data is often unavailable for many ISAs, hindering effective detection. Furthermore, collecting and labeling malware to build comprehensive datasets for each ISA is both labor-intensive and time-consuming. To reduce the burden of data collection, our idea is to translate a binary from one ISA to another ISA with sufficient malware samples. We refer to the former ISA as the ***source*** ISA, and the latter one as the ***target*** ISA. Subsequently, we can use a malware detection model trained on the target ISA to test the translated binary.

**Overview.** Figure 1 shows the workflow of applying our binary translation model, called MALTRANS, to detect malware across ISAs. Note that our goal is to reduce the effort required to collect ***task-specific datasets***—which contain labeled malware samples (as related to the step ②in Figure 1)—for training the malware detection model. This contrasts with mono-architecture datasets that can be conveniently created using open-source programs (as related to the step ①) for training MALTRANS.

In step ①, we use widely available open-source programs to train MALTRANS for translating binaries from the source ISA to the target ISA (such as X86-64). As the training is unsupervised, we cross-compile the open-source programs on different ISAs using cross-compilers to build mono-architecture datasets. Notably, *the training of* MALTRANS *does not require any malware samples*. Moreover, it should be highlighted that malware is typically a closed-source program, where the source code is usually unavailable. Thus, *cross-compilation that generates binaries across ISAs from source code does not apply to malware*.

In step ②, we train a deep learning-based malware detection model using a task-specific dataset containing malware and benign samples in the target ISA.

Finally, in step ③, when dealing with a binary in the source ISA, we use MALTRANS to translate the binary to the target ISA and reuse the malware detection model trained on the target ISA to test the translated code for detecting malware.

In summary, our approach leverages the abundance of malware samples available for the target ISA to enhance detection capabilities for other ISAs, thereby reducing the burden of data collection for malware detection. More importantly, collecting sufficient malware samples for certain ISAs can be particularly challenging. Our approach overcomes this difficulty, making robust malware detection feasible even for ISAs where malware collection is difficult.

## 4 MODEL DESIGN

We present the design and training of MALTRANS. Notably, the training of MALTRANS requires only mono-architecture datasets for each involved ISA, without the need for any malware samples.

### 4.1 INSTRUCTION NORMALIZATION

A binary, after being disassembled, is expressed in an assembly language. Given this insight, a surge of NLP-inspired binary analysis approaches have been proposed Li et al. (2022; 2023); Zan et al. (2022); Ye et al. (2023). A binary is represented as a sequence of instructions. An instruction includes an opcode (specifying the operation to be performed) and zero or more operands (specifying registers, memory locations, or literals). For example, mov eax, ebx is an instruction where mov is an opcode and eax and ebx are operands.

In NLP, the out-of-vocabulary (OOV) issue is a well-known problem, and it exacerbates significantly in our case, as constants, strings, and address offsets are frequently used in instructions. To mitigate the OOV problem, we employ the normalization strategy. Furthermore, according to a study by Jean et al. (2015), learning a translation model with a large vocabulary can significantly increase the computation complexity and hamper translation performance. Thus, normalizing instructions can also reduce the vocabulary size of both source ISA and target ISA, as well as minimize the vocabulary size discrepancy, thereby enhancing translation performance (see the evaluation).

Below are the normalization rules. Appendix A shows examples by the application of these rules.

- (R1): We use IDA Pro IDA (2023) to disassemble binaries, which generates dummy names Dummy name (2023). We replace dummy names with their respective prefixes. E.g., i) word_, dword_, and xmmword_ represent data of different lengths. They are replaced with <WORD>, <DWORD> and <XMMWORD>. ii) off_ and seg_ represent offset pointer value and segment address value. They are replaced with <OFF> and <SEG>. Other symbols are replaced with <TAG>.
- (R2): Function names are replaced with <FUNC>.
- (R3): Number constants are replaced with <VALUE>. Hexadecimal numbers are replaced with <HEX>. Minus signs are preserved.

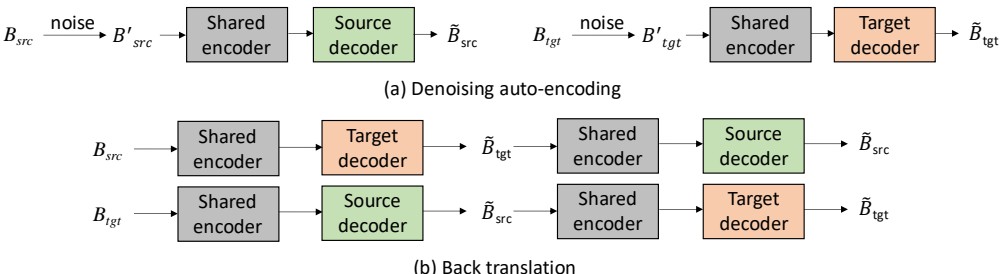

Figure 2: Training MALTRANS on the denosing auto-encoding and back-translation objectives.

## 4.2 MODEL ARCHITECTURE

MALTRANS contains a shared encoder for both source and target ISAs and a separate decoder for each ISA. We use a multi-layer bidirectional Transformer to design the encoder and decoders. The model architecture is shown in Figure 2. We regard *opcodes/operands as words* and *basic blocks as sentences*. A basic block is a straight-line sequence of instructions with no branches inside it.

We train MALTRANS in an unsupervised manner. Pretraining is a key ingredient of unsupervised neural machine translation Conneau et al. (2020); Conneau & Lample (2019). Studies have shown that the pretrained cross-lingual word embeddings has a significant impact on the performance of an unsupervised machine translation model Yang et al. (2018). We adopt this and pretrain the encoder and decoders of MALTRANS to bootstrap the iterative process of our binary translation model.

## 4.3 MODEL PRETRAINING

We employ the causal language modeling (CLM) and masked language modeling (MLM) objectives to train the encoder and decoder. (1) The CLM objective involves training the model to predict a token $e_t$, given the previous $(t - 1)$ tokens in a basic block $P(e_t|e_1, ..., e_{t-1}, \theta)$. (2) For MLM, we randomly sample $15\%$ of the tokens within the input basic block and replace them with [MASK] $80\%$ of the time, with a random token $10\%$ of the time, or leave them unchanged $10\%$ of the time.

The first and last token of an input basic block is a special token [/s], which marks the start and end of a basic block. We add position embedding and architecture embedding to token embedding, and use this combined vector as the input to the bi-directional Transformer network. Position embeddings represent different positions in a basic block, while architecture embeddings specify the architecture of a basic block. Both position and architecture embeddings are trained along with the token embeddings and help dynamically adjust the token embeddings based on their locations.

## 4.4 MODEL TRAINING FOR CODE TRANSLATION

We train MALTRANS in an unsupervised manner using the following learning objectives: denoising auto-encoding (DAE) and back translation (BT).

**Denoising Auto-Encoding (DAE).** The DAE reconstructs a basic block from its noised version, as depicted in Figure 2(a). Given the input source block, $B_{src}$, we introduce random noise into it (e.g., altering the token order by making random swaps of tokens), resulting in the noised version, $B'_{src}$. Then, $B'_{src}$ is fed into the shared encoder, whose output is analyzed by the source decoder. The training aims to optimize both the shared encoder and source decoder to effectively recover $B_{src}$. Through this, the model can better accommodate the inherent token order divergences. Similarly, the shared encoder and source decoder are optimized when the input is a target basic block, $B_{tgt}$.

**Back Translation (BT).** We adapt the back-translation approach Feldman & Coto-Solano (2020) to train our model in a translation setting, as shown in Figure 2(b). As an example, given an input source block $B_{src}$, we use the model to translate it to the target ISA (i.e., applying the shared encoder and the target decoder), as shown in the process ③. We then feed the translated block $\widetilde{B}_{tgt}$ to the model and train it to predict the original block $B_{src}$ (i.e., applying the shared encoder and the source decoder), as shown in the process ④. As training progresses, the model produces better synthetic basic block pairs through back translation, which serve to further improve the model in the subsequent iterations.

After training, MALTRANS is able to translate basic blocks across ISAs. During testing, it translates each basic block of a given binary and concatenates the translated blocks into a new translated binary.

## 5 EVALUATION

### 5.1 EXPERIMENTAL SETTINGS

We implemented MALTRANS using Transformer. Specifically, for each shared encoder and separate decoder, we used a 4-layer Transformer with 64 hidden units, 4 heads, ReLU activations, a dropout rate of 0.1, and learned positional embeddings. Appendix C presents the details of the model. All the experiments were conducted on a computer with a 64-bit 3.6 GHz Intel Core i9-CPU, a Nvidia GeForce RTX 4090, 64GB RAM, and 4TB HD.

**Model Comparison.** We consider two types of models for comparison as follows.

- *Baseline Model 1:* UnsuperBinTrans Ahmad & Luo (2023). To the best of our knowledge, UnsuperBinTrans is the first and only existing work that focuses on binary code translation using deep learning techniques. UnsuperBinTrans employs an encoder-decoder architecture, using RNNs for the encoder and decoder, and is based on unsupervised training.
- *Baseline Model 2: IR-based malware detection model.* Intermediate representation (IR) can abstract away architecture differences inherent in different ISAs, and represent instruction sets of different ISAs in a uniform style angr (2024). We consider a malware detection model trained on IR code as a baseline.
- *Optimal Model: Same-ISA model.* We consider a malware detection model trained and tested on the *same* ISA, *without employing any translation*, as the optimal model. As expected, this model, *if trained with sufficient data*, is likely to outperform a model trained on one ISA and tested on another, representing the best-case scenario. Moreover, *if the performance difference between the best model and our model is small, it indicates effective translation.*

### 5.2 TRAINING MALTRANS

We consider six ISAs: X86-64, i386, ARM64, ARM32, MIPS32, and s390x. Our goal is to reuse the malware detection model trained on X86-64 (where sufficient malware samples are available) for the other four ISAs. To achieve this, we translate binaries from the other five ISAs to X86-64.

**Mono-Arch Datasets for Training MALTRANS.** We first collect various open-source programs, including *openssl-1.1.1p*, *binutils-2.34*, *findutils-4.8.0*, and *libgpg-error-1.45*. They are widely used in prior NLP-based binary analysis works Ding et al. (2019); Li et al. (2021). We compile these programs on each ISA using GCC-11.4.0 with different optimization levels (O0-O3). We then disassemble each binary using IDA Pro IDA (2023) and collect basic blocks, which are normalized and deduplicated. Finally, we create a mono-architecture dataset for each ISA: 2,789,119 blocks for X86-64, 2,803,557 blocks for i386, 7,413,083 blocks for ARM64, 5,812,795 blocks for ARM32, 4,813,685 blocks for MIPS32, and 5,365,474 blocks for s390x. We evaluate the adequacy of our datasets, as detailed in Appendix B.

Note that the datasets used for training MALTRANS have *no overlap* with (1) the dataset used for testing the translation capability of MALTRANS and (2) the testing dataset used in the malware detection task. The details of these datasets are introduced in the following sections.

**Byte Pair Encoding (BPE).** After creating the mono-arch dataset for each ISA, we use byte pair encoding (BPE) (Sennrich et al., 2016) to process the datasets. The BPE merge times can change the vocabulary size. Based on our investigations, we find that the vocabulary size discrepancy between the source and target ISA plays a critical role in the model's translation performance. Therefore, to enhance MALTRANS's translation capability, we set the BPE merge times based on the following principles derived from empirical experiments and theoretical insights presented in prior research:

- The vocabulary size discrepancy between the source and target ISA should not exceed 15%. A large vocabulary discrepancy can lead to an imbalanced learning problem, where the model disproportionately focuses on the larger vocabulary, resulting in inefficiencies or overfitting to the less-represented vocabulary Gowda & May (2020).

Table 1: Vocabulary size, BPE merge times, and joint vocabulary size for each pair of ISAs.

| ISA Pair (src ↔ tgt) | Vocab. Size (src) | Vocab. Size (tgt) | Merge Time | Joint Vocab. Size |
|---|---|---|---|---|
| i386 ↔ X86-64 | 7,135 | 7,104 | 10,000 | 9,688 |
| ARM32 ↔ X86-64 | 9,416 | 9,236 | 22,000 | 17,262 |
| ARM64 ↔ X86-64 | 5,142 | 4,455 | 9,000 | 7,104 |
| MIPS32 ↔ X86-64 | 10,995 | 11,620 | 14,000 | 12,685 |
| s390x ↔ X86-64 | 7,148 | 6,484 | 9,000 | 8,386 |

Table 2: BLEU scores of MALTRANS and the baseline.

| ISA Pair (src → tgt) | MALTRANS (*our work*) | UnsuperBinTrans (*baseline*) |
|---|---|---|
| i386 → X86-64 | 0.40 | 0.35 |
| ARM32 → X86-64 | 0.32 | 0.28 |
| ARM64 → X86-64 | 0.34 | 0.32 |
| MIPS32 ↔ X86-64 | 0.35 | 0.27 |
| s390x → X86-64 | 0.37 | 0.29 |

- The vocabulary size of each ISA should $< 12k$ to balance capturing word semantics with computational resource constraints. A large vocabulary sizes can negatively impact model performance due to increased complexity and sparse token distributions Jean et al. (2014).

Table 1 shows the BPE merge times, vocabulary size of each ISA, and the joint vocabulary size of each ISA pair. We can see that the vocabulary discrepancies across the three ISA pairs are small, making them well-suited for training. Note that these principles are tailored to our specific scenarios.

**Training Details.** We first pre-train the encoder and decoder using the CLM and MLM tasks for the initial 2000 epochs. This helps the model learn semantic properties and contextual representations of a single ISA. Next, we train the model on the DAE and back-translation tasks, enabling it to understand code semantics across ISAs. The training continues until the evaluation loss drops below 0.5. The encoder has 788,190 parameters, while the source decoder and target decoder have 855,262 parameters each. The training time takes around $23h$, $22h$, $25h$, $23h$, and $22h$, for i385↔X86-64, ARM32↔X86-64, ARM64↔X86-64, MIPS32↔X86-64, and s390x↔X86-64, respectively.

## 5.3 TESTING MALTRANS

We use the Bilingual Evaluation Understudy (BLEU) Papineni et al. (2002) score, which is commonly used to evaluate the quality of machine-generated translations by measuring the $n$-gram overlap between the translation and the reference. We set the tokenization of SacreBLEU Post (2018) to None, apply add-one smoothing, and use the default settings to compute the BLEU score as the average precision of unigrams, bigrams, trigrams, and 4-grams.

**Mono-Arch Datasets for Testing MALTRANS.** We use three packages, *zlib-1.2.11*, *coreutils-9.0*, and *diffutils-3.7*, to test MALTRANS. Note that *these packages are not included in the training dataset of* MALTRANS. We compile them on the six ISAs using GCC-11.4.0 with different optimization levels (O0-O3) and use IDA Pro to disassemble them.

**Translation Results.** We consider five ISAs, i386, ARM32, ARM64, MIPS32, and s390x, as the source ISAs, and X86-64 as the target ISA. For each binary $B_1$ in the source ISA, there exists a semantically equivalent binary $B_2$ in X86-64, provided they stem from the same piece of source code. We use MALTRANSto translate $B_1$ into X86-64, resulting in a translated binary $B_3$ in X86-64. The BLEU score is then computed between the translated binary $B_3$ and the reference X86-64 binary $B_2$. We report the average BLEU score for all binaries. The results are shown in Table 2.

We compare MALTRANS to the baseline UnsuperBinTrans. We use the open-source trained model of UnsuperBinTrans for this comparison. Note that UnsuperBinTrans focuses solely on two ISAs (X86-64 and ARM32). To ensure a comprehensive comparison, we train UnsuperBinTrans on the same mono-architecture training datasets for the additional ISA pairs. We can see that MALTRANS outperforms UnsuperBinTrans across all ISA pairs and demonstrates satisfactory performance. Thus, MALTRANS has good translation quality and can effectively translate binaries across ISAs.

The average time to translate a basic block from one ISA to x86-64 is $10^{-4}$s, and the average number of basic blocks in a binary is $12,000$. Therefore, translating a binary takes approximately 1.2s.

Table 3: Malware detection results. We compare the detection performance by translating malware using MALTRANS and UnsuperBinTrans, and evaluate it against the IR-based and optimal model.

(a) MALTRANS vs. Two baselines.

| ISA Pair | MALTRANS | | UnsuperBinTrans | | IR-based Model | |
| (src → tgt) | AUC | F1 | AUC | F1 | AUC | F1 |
| --- | --- | --- | --- | --- | --- | --- |
| i386 → X86-64 | **0.996** | **0.996** | 0.723 | 0.649 | 0.819 | 0.805 |
| ARM32 → X86-64 | **0.981** | **0.972** | 0.818 | 0.830 | 0.815 | 0.829 |
| ARM64 → X86-64 | **0.915** | **0.904** | 0.638 | 0.594 | 0.825 | 0.749 |
| MIPS32 → X86-64 | 0.920 | 0.917 | 0.725 | 0.733 | 0.791 | 0.785 |
| s390x → X86-64 | **0.923** | **0.912** | 0.689 | 0.674 | 0.476 | 0.561 |

(b) The optimal model.

| Train & Test on the *Same* ISA | Optimal Model | |
| | AUC | F1 |
| --- | --- | --- |
| i386 | 0.998 | 0.995 |
| ARM32 | 0.986 | 0.983 |
| ARM64 | 0.705 | 0.684 |
| MIPS32 | 0.952 | 0.939 |
| s390x | 0.650 | 0.662 |

## 5.4 MALWARE DETECTION TASK

We apply MALTRANS to the malware detection task[1]. We first train a malware detection model on X86-64. When handling a binary in a given ISA, we translate it to X86-64 using MALTRANS and reuse the model trained on X86-64 to test the translated code.

**Malware Detection Model.** We use the Long Short Term Memory (LSTM) model proposed in HaddadPajouh et al. (2018) to detect malware. We design LSTM as two layers. Appendix D presents the details of the model. We first extract the token embeddings from MALTRANS, and integrate the token embeddings into the input layer of LSTM. As a result, when feeding a binary into LSTM, each input token is represented as its corresponding embedding. To further enrich the malware detection task, we also apply a Convolutional Neural Network (CNN) model. In the following, we focus on the results related to LSTM, while the results for CNN are provided in Appendix E.

**Task-Specific *Training* Datasets.** We first collect 2140, 1740, 1581, 58, 1430 and 2 malware samples from *VirusShare.com* VirusShare (2020) for X86-64, i386, ARM32, ARM64, MIPS32, and s390x, respectively. Since the optimal model trains and tests a malware detection model on the *same* ISA, it requires task-specific training data in each ISA. We spend significant efforts in collecting malware, particularly for ARM64 and s390x. It is worth noting that our approach only needs malware in a high-resource ISA for training a detection model and can greatly save the efforts in data collection.

For each ISA, we build its training and testing dataset. We divide the malware samples in each ISA into two parts: 80% are used for training and 20% for testing (for s390x, 1 malware sample is used for training and 1 for testing). In each training and testing dataset, we also include an equal number of benign programs. In the training dataset, the benign samples are randomly selected from *openssl-1.1.1p*, *binutils-2.34*, *findutils-4.8.0*, and *libgpg-error-1.45*.

**Task-Specific *Testing* Datasets.** The testing dataset for each ISA includes equal numbers of malware and benign samples. The benign samples are randomly selected from *zlib-1.2.11*, *coreutils-9.0*, and *diffutils-3.7*. Note that *these programs are not included in the dataset for training* MALTRANS *or the dataset for training the malware detection model*.

For s390x, due to the limited availability of malware, we create an imbalanced testing dataset with 1 malware and 100 benign samples. Although the dataset is limited, it reflects the value of MALTRANS: by translating code from a low-resource ISA to a high-resource ISA, we can leverage a model trained on the high-resource ISA, addressing data scarcity that would otherwise hinder robust detection for low-resource ISAs. We use AUC and F1-score as our evaluation metrics. We draw the Receiver Operating Characteristic (ROC) Curve and calculate the Area Under ROC (AUC) score as an evaluation metric. The F1-score is obtained by computing the harmonic mean of precision and recall(particularly suited for imbalanced datasets)

**Result Analysis.** We first train LSTM on X86-64, and reuse the model to test binaries in the other ISAs. The results are shown in Table 3. We can see that when the model trained on X86-64 is transferred to i386, ARM32, ARM64, MIPS32, and s390x, it achieves AUC = 0.996, 0.981, 0.915, 0.920 and 0.923, respectively. The high accuracies demonstrate the superior translation quality of MALTRANS. It should be noted that the s390x results primarily demonstrate MALTRANS's adaptability in low-resource conditions. Future work will focus on including a larger s390x dataset

---

[1]Malware packing is used to hide malicious code within benign files. In this work, we do not consider malware packing as it falls outside the scope. All malware samples are unpacked and can be analyzed by reverse-engineered tools. Appendix H discusses how to handle packed malware.

for a more comprehensive evaluation. Appendix F presents the results where the LSTM model is trained on X86-64 and tested on *all* malware samples in the other ISAs.

We then analyze how MALTRANS is able to preserve the code semantics through translation. Specifically, we visualize the embeddings of opcode tokens from different ISAs. We take the X86-64 and ARM32 pair as an example. Opcodes, which determine the operation to be performed, capture more semantics compared to operands, so we focus on opcodes for this demonstration. We extract the embeddings of 138 X86-64 opcodes and 247 ARM32 opcodes from MALTRANS, and visualize them using t-SNE, as shown in Figure 3. Four categories of opcodes are selected for demonstration. We can see that opcodes

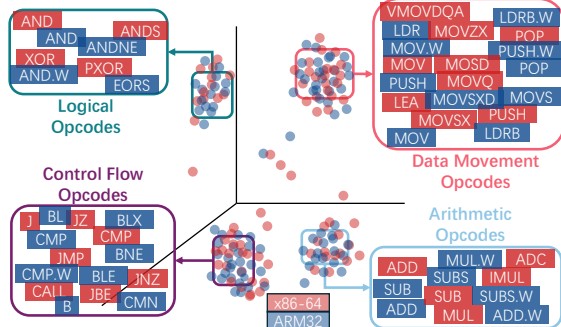

Figure 3: Visualization of opcode embeddings.

performing similar operations, *regardless of their ISAs*, are close together. Thus, MALTRANS can successfully capture semantic relationships of opcodes across ISAs and preserve code semantics.

## 5.5 MODEL COMPARISON

We compare MALTRANS to the two baselines and optimal model (as described in Section 5.1).

**Comparison with Baseline Methods.** The results are shown in Table 3(a). The first baseline is UnsuperBinTrans. As UnsuperBinTrans focuses solely on two ISAs, we train it on the same training datasets for the additional ISA pairs. We use UnsuperBinTrans to translate binaries from the other ISAs to X86-64, and use LSTM to test the translated binaries. The results show that MALTRANS achieves better translation quality, leading to improved malware detection performance.

The second baseline analyzes IR code. We assess whether a model trained on X86-64 IR code can be reused to test IR code in other ISAs. We use angr angr (2024) to lift binaries into IR code. We train LSTM using the same task-specific training dataset in X86-64, and apply the model to test the same task-specific datasets in the other ISAs. The AUC scores are 0.819, 0.815, 0.825, 0.791, and 0.476 for i386, ARM32, ARM64, MIPS32, and s390x, respectively. This indicates that IR alone does not magically allow a model trained on one ISA to be effectively reused across different ISAs.

**Comparison with Optimal Model.** The results of the optimal model are shown in Table 3(b). When the LSTM model is trained and tested on the *same* ISA, it achieves AUC of 0.998, 0.986, 0.705, 0.952, and 0.650 for i386, ARM32, ARM64, MIPS32, and s390x, respectively. Comparing the results to those of our model in Table 3(a), we observe: (1) our model achieves performance very close to the optimal model for testing malware in i386, ARM32, and MIPS32, and (2) our model significantly outperforms the optimal model for ARM64 and s390x.

For i386, ARM32, and MIPS32, the results align with expectations: the optimal model trained and tested on the *same* ISA outperforms our model, which is trained on X86-64 and tested on other ISAs (through translation). However, for ARM64 and s390x, our model outperforms the optimal model, due to the insufficient malware samples used to train the optimal model. For ARM64, only 80% of 58 ARM64 malware samples are used for training, while for s390x, only 1 out of 2 malware samples is used. This highlights the importance of a sufficiently large training dataset to achieve desirable performance. While increasing the dataset size could enhance the model's performance, collecting malware samples for certain ISAs can be challenging. Our approach—translating binaries to x86-64—addresses this data collection challenge effectively. In an extreme case, *if only one binary in a given ISA is available, we can still detect whether it is malware using the X86-64-trained model.*

## 5.6 HYPERPARAMETER AND ABLATION STUDY

**Normalization Rules.** Each instruction in the datasets is normalized by applying the three rules (R1, R2, and R3) discussed in Section 4.1. Normalization is a vital step in our approach. In this experiment, we conduct ablation study by removing certain rules and evaluating their influence on malware detection. We consider these cases:

Table 4: Impacts of normalization rules.

| Case | i386→X86-64 | | ARM32→X86-64 | | ARM64→X86-64 | | MIPS32→X86-64 | | s390x→X86-64 | |
|------|------|------|------|------|------|------|------|------|------|------|
| | AUC | F1 | AUC | F1 | AUC | F1 | AUC | F1 | AUC | F1 |
| C1 | **0.996** | **0.996** | **0.981** | **0.972** | **0.915** | **0.904** | **0.920** | **0.917** | **0.923** | **0.912** |
| C2 | 0.862 | 0.851 | 0.416 | 0.375 | 0.543 | 0.521 | 0.624 | 0.617 | 0.623 | 0.617 |
| C3 | 0.883 | 0.849 | 0.422 | 0.360 | 0.566 | 0.575 | 0.617 | 0.601 | 0.570 | 0.503 |
| C4 | 0.827 | 0.774 | 0.591 | 0.472 | 0.526 | 0.533 | 0.613 | 0.591 | 0.422 | 0.451 |
| C5 | 0.793 | 0.751 | 0.339 | 0.342 | 0.436 | 0.453 | 0.523 | 0.510 | 0.422 | 0.473 |

Table 5: AUC values when varying embedding dimension.

| ISA Pair (src → tgt) | Dimension: 32 | Dimension: 64 | Dimension: 128 |
|------|------|------|------|
| i386→X86-64 | 0.995 | 0.996 | 0.994 |
| ARM32→X86-64 | 0.978 | 0.981 | 0.972 |
| MIPS32→X86-64 | 0.912 | 0.920 | 0.869 |
| ARM64→X86-64 | 0.857 | 0.915 | 0.860 |
| s390x→X86-64 | 0.832 | 0.923 | 0.917 |

- (**C1**): Applying all rules to the data.
- (**C2**): Removing R1, and applying R2 and R3 to the data.
- (**C3**): Removing R2, and applying R1 and R3 to the data.
- (**C4**): Removing R3, and applying R1 and R2 to the data.
- (**C5**): Not applying any rules to the data.

Table 4 shows the results. We can observe that: (1) When all normalization rules are applied (**C1**), we achieve the best performance. (2) When a subset of normalization rules is applied (**C2-4**), the AUC values are lower than in **C1**, indicating that each normalization rule mitigates the OOV issue and has an impact on translation quality, thereby influencing malware detection performance. (3) When no normalization rule is applied (**C5**), the results are the lowest.

**Embedding Dimension.** We evaluate the impacts of the embedding dimension. We test different dimension sizes, including 32, 64, and 128, to train MALTRANS. We then apply MALTRANS to translate binaries from the source ISA to X86-64 for malware detection. The results are shown in Table 5. We observe that when the dimension is set to 64, the AUC values are higher compared to other dimensions. Moreover, as the dimension increases, the training time also increases. We thus choose a dimension of 64, considering both the translation quality and training efficiency.

**Summary.** These results highlight the significant advantages of MALTRANS in malware detection. (1) It eliminates the need for task-specific data (i.e., malware samples) in the source ISA to train a malware detection model, thereby reducing the burden of data collection. (2) It enables malware detection in the source ISA using a model trained on the target ISA and achieves high detection accuracies. (3) Training the binary translation model MALTRANS does not require malware samples. Instead, it only requires mono-architecture datasets, which can be easily and conveniently created by cross-compiling open-source programs using cross-compilation tools like QEMU QEMU (2023) and LLVM LLVM (2023). Thus, our approach is highly convenient and feasible.

## 6 CONCLUSION

Applying deep learning to malware detection has drawn great attention. The limited availability of malware in certain ISAs, however, hinders deep learning-based malware detection. In this work, we proposed MALTRANS, which translates binaries across ISAs. The training of MALTRANS does not require malware and relies only on mono-architecture datasets created from open-source programs. We apply MALTRANS to malware detection across six ISAs. Considering that X86-64 is the most data-rich ISA, we train MALTRANS to translate binaries from the other five ISAs to X86-64, and reuse a malware detection model trained on X86-64 to test the translated code. Our approach effectively reduces the burden of data collection. Compared to UnsuperBinTrans, MALTRANSachieves better malware detection across ISAs, thanks to its improved model architecture and tailored normalization schemes, contributing to its superior translation capability. Furthermore, we expanded the evaluation to cover more ISAs, showcasing MALTRANS's broader applicability and improved performance.

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

## A APPLYING INSTRUCTION NORMALIZATION RULES

In Section 4.1, we outlines the instruction normalizing rules (R1)-(R3). Below, we provide examples demonstrating how these rules are applied to assembly code from six ISAs, as shown in Table 6. For each ISA, the assembly code on the left side of the sub-table represents the original code, while the code on the right side represents the normalized version.

Table 6: Comparison of original and normalized assembly code.

```
call _gpgrt_log_info       call <FUNC>              add esp 0Ch                   add esp <HEX>
sbb al 0                   sbb al <VALUE>           call _dcgettext              call <FUNC>
mov esi 0ACh+_bss_start    mov esi <HEX>+<TAG>      lea eax [ebx-5D40h]          lea eax [ebx-<HEX>]
lea rdi str_LogWithPid     lea rdi <STR>            jmp short loc_37C3           jmp short <LOC>
sub rdx buffer             sub rdx <TAG>            mov eax [150h+domainname]    mov eax [<HEX>+<TAG>]
```
              (a) X86-64                                                    (b) i386

```
ADD R12 R12 0x1B000        ADD R12 R12 <HEX>        LDR X0 [SP #0xC0+stream_68]  LDR X0 [SP <HEX>+<TAG>]
LDR PC memcpy-0x2B7C       LDR PC <FUNC>-<HEX>      TBZ W0 #0 loc_AF38           TBZ W0 <HEX> <LOC>
BEQ.W loc_109B4            BEQ.W <LOC>             ADRL X1 str_ErrorInitia       ADRL X1 <STR>
BL gz_uncompress           BL <FUNC>               B _gmon_start_               B <FUNC>
CMP R2 0                   CMP R2 <VALUE>           MOV X19 #0                   MOV X19 <HEX>
```
              (c) ARM32                                                     (d) ARM64

```
lw $fp 0x40+var_20         lw $fp <HEX>+       lgf %r1 0xC(%r1)             lgf %r1 <HEX>(%r1)
bal usage                  bal <FUNC>              larl %r1 _ctype_b_loc_ptr    larl %r1 <TAG>
beqz $v0 loc_1358          beqz $v0 <LOC>          jne LOC_E84                  jne <LOC>
move $a2 $s3+1             move $a2 $s3+<VALUE>     stg %r1 arg_190              stg %r1 <ARG>
sw $s0 0x40+path($sp)      sw $s0 <HEX>+<TAG>($sp)  sllg %r1 %r1 1               sllg %r1 %r1 <VALUE>
```
              (e) MIPS32                                                    (f) s390x

## B DATASET ADEQUACY

In NLP, it is widely recognized that a comprehensive dataset, which ensures that the vocabulary covers a wide range of words, is crucial for training effective code translation models. We assess the adequacy of our mono-architecture datasets. Specifically, we study the vocabulary growth as we incrementally include programs. Our findings indicate that while the vocabulary size initially increases with the inclusion of more programs, it eventually stabilizes. Take x86-64 as an example, including *openssl-1.1.1p* results in a vocabulary size of $23,029$. The size increases to $36,770$ (a $60\%$ growth) when *binutils-2.34* is added, and then increases to $39,499$ (a $7.4\%$ growth) and $39,892$ (a $0.9\%$ growth) when *findutils-4.8.0*, *libpgp-error-1.45* are included, respectively. The growth trend is similar for other ISAs. It shows that the vocabulary barely grows in the end when more programs are added. According to the vocabulary growth trend as well as the high performance achieved, our mono-architecture datasets are adequate to cover instructions and enable effective code translation.

## C MODEL PARAMETERS OF MALTRANS MODEL

The shared encoder and separate decoders of MALTRANS are implemented using the Transformer model. Table 7 shows the details of the parameters.

## D MODEL PARAMETERS OF MALWARE DETECTION LSTM MODEL

We use the LSTM model to detect malware. Table 8 shows the parameters details of the LSTM model.

## E MALWARE DETECTION USING A CNN MODEL

We use a 1-dimensional Convolutional Neural Network (CNN) model to detect malware. The parameter details of the CNN model are presented in Table 9.

We use the same task-specific training and testing datasets described in Section 5.4. Moreover, we compare our results against two baseline methods and the optimal model. Table 10 presents the results of the malware detection task using the CNN model.

Table 7: Parameter Details of MALTRANS.

| Parameter | Value | Description |
|---|---|---|
| Emb. Dimension | 32/64/128 | Embedding layer size for tokens |
| Hidden Dimension | 4* Emb. Dimension | Transformer FFN hidden dimension |
| Num. Layers | 4 | Number of transformer layers |
| Num. Heads | 4 | Number of attention heads per layer |
| Regu. Dropout | 0.1 | Dropout rate for regularization |
| Attn. Dropout | 0.1 | Dropout rate in attention layers |
| Batch Size | 256 | Number of sentences per batch |
| Max. Length | 512 | Maximum length of one sentence after BPE |
| Optimizer | Adam | Adam optimizer with sqrt decay |
| Clip Grad. Norm | 5 | Maximum gradient norm for clipping |
| Act. Function | ReLU | Use ReLU for activation |
| Pooling | Mean | Use mean pooling for sentence embeddings |
| Accumulate Grad. | 8 | Accumulate gradients over N iterations |

Table 8: Parameter Details of the malware detection LSTM model.

| Parameter | Value | Description |
|---|---|---|
| Emb. Dimension | 32/64/128 | Input feature dimension for sequence embedding |
| Num. of Layers | 3 | Number of stacked LSTM layers in the network |
| Hidden Units | 16 | Number of hidden units in each LSTM layer |
| Output Units | 1 | Dimension of the output layer |
| Batch Size | 36 | Number of samples processed in one batch |
| Optimizer | Adam | Adaptive optimization algorithm with momentum |
| Loss Function | BCEWithLogitsLoss | Binary cross-entropy with logits |
| Pooling | Max | Maximum value across temporal dimension |

Table 9: Parameter Details of the malware detection CNN model.

| Parameter | Value | Description |
|---|---|---|
| Emb. Dimension | 64 | Input feature dimension for sequence embedding |
| Conv. Layers | 2 | Number of convolutional layers in the CNN network |
| Conv. Kernel | 3 | Kernel size of the convolutional layers |
| Output Units | 1 | Dimension of the output layer |
| Batch Size | 64 | Number of samples processed in one batch |
| Optimizer | Adam | Adam optimizer with a learning rate of 0.001 |
| Loss Function | BCEWithLogitsLoss | Binary cross-entropy with logits |
| Pooling | Max | Maximum pooling to reduce spatial dimensions |

Table 10: Malware detection results using the CNN model. We compare the detection performance by translating malware using MALTRANS and UnsuperBinTrans, and evaluate it against the IR-based model and the optimal model.

(a) MALTRANS vs. Two baselines.

| ISA Pair | MALTRANS | | UnsuperBinTrans | | IR-based Model | |
|---|---|---|---|---|---|---|
| (src → tgt) | AUC | F1 | AUC | F1 | AUC | F1 |
| i386 → X86-64 | **0.992** | **0.989** | 0.693 | 0.630 | 0.724 | 0.706 |
| ARM32 → X86-64 | **0.973** | **0.962** | 0.769 | 0.726 | 0.718 | 0.730 |
| ARM64 → X86-64 | **0.919** | **0.902** | 0.645 | 0.674 | 0.725 | 0.789 |
| MIPS32 → X86-64 | 0.924 | 0.917 | 0.805 | 0.673 | 0.761 | 0.745 |
| s390x → X86-64 | **0.922** | **0.910** | 0.606 | 0.648 | 0.486 | 0.507 |

(b) The optimal model.

| Train & Test on the *Same* ISA | Optimal Model | |
|---|---|---|
| | AUC | F1 |
| i386 | 0.995 | 0.963 |
| ARM32 | 0.976 | 0.940 |
| ARM64 | 0.738 | 0.691 |
| MIPS32 | 0.955 | 0.941 |
| s390x | 0.570 | 0.630 |

When the CNN model trained on X86-64 is transferred to i386, ARM32, ARM64, MIPS32, and s390x, it achieves AUC values of 0.992, 0.973, 0.919, 0.924, and 0.922, respectively. These high accuracies highlight the superior translation quality of MALTRANS, outperforming both UnsuperBinTrans and the IR-based model. Compared to the optimal model, we have the following observation. (1) First, our model achieves performance close to the optimal model when testing malware on i386, ARM32,

and MIPS32. This outcome is expected, as the optimal model is trained and tested on the same ISA, while our model is trained on X86-64 and tested on other ISAs through translation. (2) Second, our model significantly outperforms the optimal model on ARM64 and s390x. This is due to the limited malware samples available for training the optimal model on the two ISAs, highlighting the value of our approach. By reusing a model trained on a high-resource ISA, we enable robust detection for low-resource ISAs that would otherwise face significant challenges.

## F  MALWARE DETECTION USING ALL MALWARE SAMPLES FOR TESTING

We train the LSTM model exclusively on X86-64, and reuse the trained model to test binaries in other ISAs, including i386, ARM32, ARM64, MIPS32, and s390x. It is important to note the key difference between the experiment described in this Appendix and that in Section 5.4. Here, we use all malware samples from i386, ARM32, ARM64, MIPS32, and s390x for testing. In contrast, in Section 5.4, only 20% of the malware samples from these ISAs are used for testing, as the remaining 80% are reserved for training the optimal model.

**Task-Specific *Training* Dataset.** We first build the training dataset containing an equal number of malware and benign samples in X86-64. We collect 2140 malware samples from *VirusShare.com* (VirusShare, 2020), where $80\%$ (=1712) are used for training and $20\%$ (=428) for testing. The benign samples are randomly selected from *openssl-1.1.1p*, *binutils-2.34*, *findutils-4.8.0*, and *libgpg-error-1.45*.

**Task-Specific *Testing* Datasets.** We build a testing dataset for each ISA. The benign samples are randomly selected from three packages: *zlib-1.2.11*, *coreutils-9.0*, and *diffutils-3.7*. Note that *these packages are not included in the dataset for training* MALTRANS *or the task-specific dataset for training the malware detection LSTM model.*

For X86-64, the testing dataset contains 428 malware samples and 428 benign samples. For the other ISAs, we collect 1740, 1581, 58, 1430, and 2 malware samples from *VirusShare.com* VirusShare (2020) for i386, ARM32, ARM64, MIPS32, and s390x, respectively. The testing datasets for i386, ARM32, ARM64, and MIPS32 contain equal numbers of malware and benign samples. However, due to the limited availability of malware for s390x, we create an imbalanced testing dataset with 2 malware samples and 100 benign samples. We report both AUC and F-1 score as the evaluation metrics.

Table 11: Malware detection results. We use all the malware samples in i386, ARM32, ARM64, MIPS32, and s390x for testing. We compare the detection performance by translating malware using MALTRANS and UnsuperBinTrans, and evaluate it against the IR-based model.

| ISA Pair (src → tgt) | MALTRANS | | UnsuperBinTrans | | IR-based Model | |
|---|---|---|---|---|---|---|
| | AUC | F1 | AUC | F1 | AUC | F1 |
| i386 → X86-64 | **0.998** | **0.997** | 0.742 | 0.670 | 0.804 | 0.832 |
| ARM32 → X86-64 | **0.978** | **0.974** | 0.813 | 0.820 | 0.819 | 0.830 |
| ARM64 → X86-64 | **0.917** | **0.909** | 0.620 | 0.589 | 0.820 | 0.743 |
| MIPS32 → X86-64 | **0.951** | **0.937** | 0.730 | 0.729 | 0.789 | 0.783 |
| s390x → X86-64 | **0.921** | **0.931** | 0.681 | 0.672 | 0.428 | 0.548 |

**Result Analysis.** We first train LSTM on X86-64, and reuse the model to test binaries in the other ISAs. The results are shown in Table 11. We can see that when the model trained on X86-64 is transferred to i386, ARM32, ARM64, MIPS32 and s390x, it achieves AUC = 0.998, 0.978, 0.913, 0.923 and 0.938, respectively. The high accuracies demonstrates the superior translation quality of MALTRANS.

## G  TRANSLATION DEMONSTRATION

Table 12 and Table 13 shows some randomly selected examples. In each example, (1) the *source ISA* is the original basic block in the source ISA, which could be i386, ARM32, ARM64, MIPS32, or s390x; (2) the *target ISA* is the basic block in the target ISA, X86-64, that is similar to the original basic block in the source ISA; and (3) the *translated ISA* is the X86-64 basic block translated from the original basic block in the source ISA by our model MALTRANS.

By comparing the translated X86-64 block with the target X86-64 block, we observe that MALTRANS (1) accurately predicts almost all opcodes, and (2) while a few operands are predicted incorrectly, these errors are reasonable. Note that an instruction consists of an opcode (which specifies the operation) and zero or more operands (which specifies registers, memory locations, or literal data). Thus, opcodes, which determine the operation to be performed, capture more semantic information compared to operands. On the other hand, different registers or memory locations can be used to store data while preserving code functionality, which reduces the significance of operands.

Consider the first example of the ISA pair i386→X86-64. In the target X86-64 basic block, the second instruction is: **add** rbp [state+<HEX>], whereas in the translated X86-64 basic block, the predicted instruction is: **add** rdx [s+<HEX>]. Here, MALTRANS successfully predicts the opcode **add**, and predicts a different register and memory cell for the operands, while preserving the functionality of the code.

Table 12: Examples for code translation.

| | | | |
|---|---|---|---|
| i386 | 1 | Source i386 | **sub** esp <VALUE> **add** edx [s+<HEX>] **mov** ebx esi **push** len **push** edx **push** [esp+<HEX>+dictionary] **call** <FUNC> **add** esp <HEX> **jmp** <LOC> |
| | | Translated X86-64 | **sub** rsp <VALUE> **add** rdx [s+<HEX>] **mov** rbx rbp **push** len **mov** r8 [rsp+<HEX>+datalen] **mov** rcx len **call** <FUNC> **jmp** <ADDR> |
| | | Target X86-64 | **sub** rsp <VALUE> **add** rbp [state+<HEX>] **mov** eax <HEX> **push** <TAG> **mov** [state+<HEX>] rdi **mov** rdi state **call** <FUNC> **jmp** short <TAG> |
| | 2 | Source i386 | **mov** edx [edi+<VALUE>] **mov** edp <VALUE> **cmp** edx <HEX> **lea** ecx [edx-<HEX>] **setnz** al **and** ecx <HEX> **lea** ecx [edx-<HEX>] |
| | | Translated X86-64 | **mov** rdx [abfd+<VALUE>] **mov** ebp <VALUE> **cmp** rdx <HEX> **lea** rcx [r1+rax+<VALUE>] **setnz** al **and** ecx <HEX> **lea** rcx [r1+rax+<VALUE>] |
| | | Target X86-64 | **mov** rdx rsp **mov** rsp <HEX> **cmp** rax rdx **lea** r8 <TAG> **setnz** a1 **and** ecx <TAG> **lea** rcx [rax+<HEX>] |
| | 3 | Source i386 | **sub** esp <VALUE> **lea** eax [esi+<VALUE>] **push** eax **push** [esp+<HEX>+buf] **call** <FUNC> **mov** edi eax **mov** c [esp+<HEX>+<TAG>] |
| | | Translated X86-64 | **sub** rsp <VALUE> **lea** rax <TAG>+<HEX> **mov** cs:<TAG> rax **mov** rax [abfd+<VALUE>] **call** <FUNC> **mov** rsi rsp **mov** rdi abfd **call** <FUNC> **add** rsp <VALUE> |
| | | Target X86-64 | **sub** rsp <HEX> **lea** eax <HEX> **mov** rax rs:<HEX> **mov** [rsp+<HEX>+] **call** <FUNC> **mov** rcx rsp **mov** rsi <VALUE> **call** <FUNC> **add** eax <HEX> |
| ARM32 | 1 | Source ARM32 | **XOR** R0 R2 **MOV** R1 R1 <VALUE> **SUBS** R3 R0 R3 **LDP** R6 R6 have **BNE** <TAG> **CMP** copy R7 |
| | | Translated X86-64 | **xor** edi edi **mov** rcx rdx **and** esi <VALUE> **sub** <TAG> <VALUE> **mov** [rax+rdx-<VALUE>] cl **mov** ecx <VALUE> |
| | | Target X86-64 | **xor** edi edi **mov** [rax+rcx-<VALUE>] di **and** edi <TAG> **sub** init curr **mov** [s+<HEX>] rdx **mov** rsp <VALUE> |
| | 2 | Source ARM32 | **LDR** R2 [R3] **MOVS** R0 <VALUE> **BNE** <LOC> **LDR** R0 [R2+<TAG>] **MOV** R1 <TAG> **BL** <FUNC> |
| | | Translated X86-64 | **mov** rsi [rsp+<HEX>+p] **movsxd** rdi eax **lea** rcx [rsp+<HEX>+id] **call** <FUNC> |
| | | Target X86-64 | **mov** rdi in **movsxd** rdi file **lea** rax [rsi+<VALUE>] **call** <FUNC> |
| | 3 | Source ARM32 | **MOV** R0 strm **MOVS** R1 <VALUE> **BL** <FUNC> **ADDS** err <VALUE> **MOVS** R2 <TAG> **LDR** R1 <HEX> **BNE** <ADDR> |
| | | Translated X86-64 | **mov** edi ebx **lea** rcx <TAG> **movsxd** rax ds:<TAG> **call** <FUNC> **add** rax rcx **lea** rdx <TAG> **jmp** <ADDR> |
| | | Target X86-64 | **mov** edi ebx **lea** rsi <HEX> **movzx** rax <TAG> **call** <FUNC> **add** eax ecx **lea** rbp <LOC> **jmp** <TAG> |
| ARM64 | 1 | Source ARM64 | **MOV** W2 <VALUE> **MOV** W0 W2 **LDP** X2 <VALUE> [SP+<HEX>+<TAG>] **LDR** X2 [SP+<HEX>+<TAG>] **LDP** X2 X3 [SP+<HEX>+<TAG>] |
| | | Translated X86-64 | **mov** qword ptr [rax] <VALUE> **mov** rax [rbp+p] **mov** rdi rax **call** <FUNC> **test** eax eax **jnz** <ADDR> |
| | | Target X86-64 | **mov** qword ptr [rcx+<HEX>] <VALUE> **mov** [rdx+<HEX>] <VALUE> **mov** [rdx+<HEX>] rcx **call** <FUNC> **test** rax rax **jnz** <TAG> |
| | 2 | Source ARM64 | **ADD** W2 W2 <VALUE> **ADD** X3 X0 X3 **SUB** W2 W2 <VALUE> **MOV** W0 <VALUE> **STRB** W0 [X1+<HEX>] **MOV** X0 X2 <VALUE> |
| | | Translated X86-64 | **mov** r1 q **mov** rsi p **movzx** ri byte ptr [p] **sub** r1 <VALUE> **add** p <HEX> **mov** rax [p] **jmp** short <LOC> |
| | | Target X86-64 | **mov** r14 rsp **mov** rsip **movzx** rax <HEX> **sub** r10 <VALUE> **add** q <HEX> **mov** r11 <VALUE> **jmp** <TAG> |
| | 3 | Source ARM64 | **LDR** W5 <VALUE> **LDR** X9 <HEX> **SUB** W0 W5 W0 **ADD** W1 W0 <VALUE> **LDP** W1 <VALUE> **SUB** W0 W5 W0 **ADD** W1 W0 <VALUE> **LDP** W1 <VALUE> **LDRB** W2 <TAG> |
| | | Translated X86-64 | **mov** rdx <HEX> **mov** [rsp+<HEX>+n] rdx **movzx** edx [r1+<HEX>] **mov** esi edx **mov** r9 [r1+<HEX>] **lea** r1 [rdx+<TAG>] |
| | | Target X86-64 | **mov** rdx [rbp+mode] **mov** ecx [rbp+fd] **movzx** rax [rbp+path] **mov** esi ecx **mov** rdi rax **mov** [rbp+gz] rax **lea** r1 [rax+<VALUE>] |

Table 13: Examples for code translation.

| | | | |
|---|---|---|---|
| MIPS32 | 1 | Source MIPS32 | **li** $t9 <VALUE> **lw** $ra <HEX>+ ($sp) **addiu** $t9 <FUNC> **b** <FUNC> **addiu** $sp <HEX> |
| | | Translated X86-64 | **mov** r8 <VALUE> **mov** rbx rsp+<HEX>+<TAG> **lea** r10 r8+<ADDR> **call** <FUNC> add rbp <VALUE> |
| | | Target X86-64 | **mov** r10 <VALUE> **mov** rax rsp+<HEX>+ **lea** r10 r10+<TAG> **jmp** <FUNC> **add** rsp <HEX> |
| | 2 | Source MIPS32 | **li** $a1 <VALUE> **addiu** $a1 <STR> **move** $a0 $s0 **la** $t9 <FUNC> **jalr** $t9 **lw** $gp <HEX>+ ($sp) |
| | | Translated X86-64 | **mov** rsi <HEX> **lea** rbp rbp+<ADDR> **mov** rdi rax **mov** r11 <FUNC> **call** <FUNC> **mov** rap rsp |
| | | Target X86-64 | **mov** rsi <VALUE> **lea** rsi rsi+<STR> **mov** rdi, rbx **mov** r10 <ADDR> **call** <FUNC> **mov** rbp rsp |
| | 3 | Source MIPS32 | **li** $a2 <VALUE> **li** $a1 <VALUE> **addiu** $a1 <STR> **move** $a0 $zero **la** $t9 <FUNC> **jalr** $t9 **lw** $gp <HEX+ ($sp) **move** $a1 $s1 **move** $a0 $v0 |
| | | Translated X86-64 | **mov** rbx <HEX> **mov** rsi <ADDR>  **lea** rsi rsi+<TAG> **xor** rsi rsi **mov** r10 r9 **call** <FUNC> **mov** rbp rap **mov** rsi r14+<TAG>+<HEX> **mov** rdi rax |
| | | Reference X86-64 | **mov** rdx <VALUE> **mov** rsi <VALUE> **lea** rsi rsi+<STR> **xor** rdi rdi **mov** r10 r11 **call** <FUNC> **mov** rbp rsp+<HEX>+ **mov** rsi r14 **mov** rdi rax |
| s390x | 1 | Source s390x | **lg** %r1 <TAG> (%r11) **ag** %r1 <TAG> **stg** %r1 <TAG> (%r11) **lg** %r1 <TAG> **br** %r1 **jg** <FUNC> **basr** %r1 %r0 **lgf** %r1 <HEX> |
| | | Translated X86-64 | **mov** rdi at1 **mov** rsp [<HEX>+] **mov** rbx rdx **lea** [r14 r12+r13] **mov** rdi [rsp+<HEX>] **call** <FUNC> **test** rbx rbx **jnz** <LOC> |
| | | Target X86-64 | **mov** rsi <TAG> **mov** rbp <VALUE> **lea** [rdx+<HEX>] **mov** rdi [rax+<VALUE>] **call** <FUNC> **test** rax rax **jnz** <TAG> |
| | 2 | Source s390x | **basr** %r0 %r1 **lgf** %r0 <VALUE> (%r1) **jg** <FUNC> **lalr** %r1 <TAG> **lg** %r1 (%r1) **br** %r1 **lghi** %r8 <LOC> |
| | | Translated X86-64 | **mov** r1 r3 **mov** rdi <HEX> **lea** <FUNC> **movsx** rax <TAG> **mov** rbx rdx **push** rbp **jmp** short <LOC> |
| | | Target X86-64 | **mov** r14 rsp **mov** rsip **movsx** rbx  **lea** <TAG> **mov** r2 r4 **mov** rbp (<TAG>+<HEX>) **mov** r11 <VALUE> **jmp** <TAG> |
| | 3 | Source s390x | **xc** %r5 <VALUE>+ **stmg** %r14 %r15 **sgr** %r1 %r2 **lay** %r15+<HEX> (%r15) **lglr** %r3 <VALUE> <LOC> **ahi** %r11 <VALUE>+<TAG> **lgfr** %r4 (<HEX>+<TAG>) |
| | | Translated X86-64 | **mov** rax +<HEX> **mov** rcx rbx **mov** ebx [ecx+<VALUE>] **mov** ebi ecx **mov** r9 [r1+<HEX>] **lea** r1 [rdx+<TAG>] |
| | | Reference X86-64 | **mov** rax [<VALUE>+<HEX>] **mov** eax  **mov** rbp [rsi+<VALUE>] **mov** eax ebx **mov** rsi rbx **mov** rax (rax+<TAG>) **lea** rsi [r2+] |

## H DISCUSSION

**Uniqueness of Our Work.** We propose an entirely different approach compared to the works in Wang et al. (2023b; 2024), which also aims to reuse models across ISAs. While their approach achieves this by learning cross-architecture instruction embeddings, our method focuses on translating binary code across ISAs. By translating code to a high-resource ISA, our approach offers several advantages. First, it allows the direct application of existing downstream models—trained on the high-resource ISA—to other ISAs through testing the translated code. In contrast, the works in Wang et al. (2023b; 2024) require retraining the model using cross-architecture instruction embeddings. Furthermore, translating code from one ISA to another assists human analysts in understanding code from unfamiliar ISAs, supporting broader applications in code comprehension.

InnerEye Zuo et al. (2018) applies neural machine translation techniques for binary code similarity comparison but *does not perform binary code translation across ISAs*. It uses two encoders from neural machine translation models, where each encoder generates an embedding for a piece of binary code of a given pairs, and measures similarity based on embedding distance. In contrast, our approach focuses on translating binary code across ISAs.

**Packed or Obfuscated Malware.** To address packed malware, we can incorporate advanced unpacking tools, such as PEiD PEiD (2008) and OllyDbg OllyDbg (2000), to first unpack the malware and then analyze the unpacked content.

The malware samples used in our study were from *VirusShare.com* VirusShare (2020), a repository that collects malware observed in the wild. It is widely recognized that such malware often employs obfuscation techniques to evade detection by antivirus systems. However, we lack ground truth regarding the specific obfuscation techniques applied to each malware sample, making it difficult to

assess resilience to specific obfuscation techniques. Identifying the obfuscation techniques used in a given malware sample is a challenging and open research problem in its own right.

Our study primarily addresses the challenge of data scarcity in low-resource ISAs by translating binaries from these ISAs to a high-resource ISA using MALTRANS. Future work could systematically explore the impact of obfuscation techniques on detection performance. A key challenge is the absence of a high-fidelity dataset mapping malware samples to their specific obfuscation techniques. Filling this gap will be a focus of our future research.

