# OpenReview forum: "MalTrans: Unsupervised Binary Code Translation with Application to Malware Detection"
_ICLR.cc/2025/Conference — Submitted to ICLR 2025_

### Official Review · Reviewer_C5ox · 2024-10-23

**Soundness:** 3
**Presentation:** 2
**Contribution:** 2
**Rating:** 5
**Confidence:** 4

**Summary:**

This paper proposes MalTrans, a malware detection model that leverages translations to parse rarely-available programs in specific ISAs.
In particular, MalTrans is firstly trained with self-supervised learning to translate from one ISA (like x86) to another one (ARM32). This is done by compiling several open-source programs to different targets (x86, ARM, etc), and train a transformer model to translate them correctly. Once this is done, programs are then ported to an ISA (like x86) which is very common, and on which it is easier to build a malware detector. The paper presents different ablation studies on how to setup the components of MalTrans, while achieving better results in terms of translation and detection rate.

**Strengths:**

**Translating between ISAs is good advancement.** Since there are a lot of devices also with "esoteric" ISAs, this tool can really enable a good first detection on malware targeting IoT.

**Interesting embedding analysis.** It is interesting to see that instructions cluster together depending on their functionality.

**Good experimental results and ablation study.** It is very important that models are followed by extensive ablation study, to show how their performance changes and why, and this paper provides rightfully.

**Weaknesses:**

**Motivation is weak.** While the authors state that IoT malware are spreading more than anything, I would like to ask them detailed reports on this statement. We know that there is an on-going issue with Windows malware (https://www.virustotal.com/gui/stats by looking at the file types) and Android malware. But IoT is a blurred terminology, and I fail to understand this alarming threat.
Hence, the motivation that we need to detect malware for IoT does not hold that much.

**Missing details on how BLEU is computed.** I imagine that the BLUE is computed on the sequence to match, but it is not really clear how it was used. Hence, I believe that the authors computed the BLEU between the predicted sequence and the ground-truth created from the available source code. Is this correct? Also, it was computed on what? Basic-block level, single-instruction level? The authors must specify these details.

**i386 is x86 (32bit).** I found confusing to read the translation from i386 to x86, and I also found very confusing the fact that the BLEU score from i386 to x86 is **only** 0.4 (which means that the sequences match only by half). The 32bit version of x86 is i386. Were the authors thinking about translating to x64 (64 bit) from i386? Also, in the paper there is written x86-64 or x86. The authors should specify whether they target the 32- or 64bit ISA.
Also, while it is possible to translate from 32bit to 64bit, it does not make sense to do the opposite.

**Missing examples.** Can the authors submit examples of their translations? This might help understand better the quality of the translation.

**Maybe submit to a security venue?** While the proposal is interesting, the content might better fit another venue closer to the security community. This is a minor comment, not considered during the review process.

**Questions:**

1. How the authors computed the BLEU score for their experiments?
2. Which specific ISAs are the authors targeting with their experiments?
3. Is it true that IoT malware are so numerous to outmatch all other domains (like Windows and Android)?

---

> ### Author Response · Authors · 2024-11-17
> **Response to Reviewer C5ox**
>
> Thank you for the insightful and valuable comments. Please find our response below.
>
> > 1. *Motivation is weak… Is it true that IoT malware are so numerous to outmatch all other domains (like Windows and Android)?*
>
> **Response:** The motivation for this work stems from the data scarcity of malware, which is a significant challenge for IoT. Due to the heterogeneity of IoT devices, a variety of ISAs are used in their development. However, this diversity has led to a lack of sufficient malware data for many ISAs, hindering effective malware detection. In contrast, Windows and Android, which primarily operate on x86 and ARM architectures, do not face the issue of data scarcity.
>
> Furthermore, malware detection for platforms like Windows and Android is a well-studied problem, with numerous mature solutions available. For IoT, however, malware detection remains underexplored, and critical issues, such as the scarcity of malware data for certain ISAs, remain unresolved.
>
> Notably, IoT malware incidents surged by 87% in 2022 compared to 2021, reaching 112.3 million cases ([link](https://www.sonicwall.com/resources/white-papers/2023-sonicwall-cyber-threat-report/gated/thank-you/asset)). This sharp rise highlights the increasing security threats facing IoT ecosystems.
>
> > 2. *How the authors computed the BLEU score for their experiments?*
>
> **Response:** For each binary $B_1$ in one ISA, there exists a semantically equivalent binary $B_2$ in x86-64, provided they stem from the same piece of source code. We use MALTRANS to translate $B_1$ into X86-64, resulting in a translated binary $B_3$ in X86-64. The BLEU score is then computed between the translated binary  $B_3$ and the reference X86-64 binary $B_2$. The BLEU score is calculated at the binary file level and incorporates an average of unigram, bigram, trigram, and 4-gram precision. We will clarify this in the revision.
>
>
> > 3. *Which specific ISAs are the authors targeting with their experiments?*
>
> **Response:** We apologize for the confusion. We translate binary code from i386 to x86-64. The term "x86" used in the paper actually refers to x86-64. We will revise the paper to eliminate this ambiguity.
>
>
> > 4. *Missing examples. Can the authors submit examples of their translations? This might help understand better the quality of the translation.*
>
> **Response:** We have included some examples of our translations in Table 6 in Appendix A. Depending on space constraints, we will include additional examples in the revised version to further demonstrate the quality of the translations.

---

### Official Review · Reviewer_yy4t · 2024-10-28

**Soundness:** 4
**Presentation:** 3
**Contribution:** 4
**Rating:** 6
**Confidence:** 4

**Summary:**

This paper proposes an unsupervised binary code translation framework called MALTRANS,
designed to facilitate malware detection across multiple Instruction Set Architectures (ISAs).
It addresses the challenge of labeled data availability by leveraging neural machine translation (NMT).
The study focuses on overcoming the limitations posed by insufficient malware datasets across various ISAs.

The proposed system, MALTRANS, translates a given source ISA to a target ISA,
such as x86-64. It features a shared encoder with separate decoders for each ISA.
The paper reports that MALTRANS achieves strong performance across five different ISAs: x86-64, i386, ARM32, ARM64, and s390x.

This paper presents an interesting approach to cross-ISA malware detection,
with clear performance improvements over prior work and a broader scope by covering multiple architectures,
including some less commonly explored. The unsupervised learning framework,
which reduces dependency on labeled malware datasets, is a practical strength.
The thorough evaluation further demonstrates the system’s robustness.

The reviewer finds the paper sound and notes that it offers clear advantages over prior work.
However, several aspects warrant further consideration before acceptance. Detailed feedback is provided below.

Additionally, the discussion of real-world challenges, such as scalability and the handling of packed or obfuscated malware, could be expanded to provide a more comprehensive perspective. Strengthening the differentiation from previous approaches would also help better position the paper within the existing research landscape.

**Strengths:**

1. Practical Relevance:

The reviewer considers the paper's application across multiple ISAs, including i386, ARM32, ARM64, s390x, and X86-64,
to be a significant strength. This breadth ensures the framework’s practical relevance,
enabling deployment across diverse platforms and making an important contribution to cross-architecture malware detection.

2. Unsupervised Learning to reduce labelled data collection burden:

The reviewer believes that the use of unsupervised learning to eliminate
the need for labeled malware datasets is a meaningful contribution in the cybersecurity domain.

3. Improved Performance:

The paper reports strong empirical results, including higher BLEU and AUC scores compared to
prior baselines such as UnsuperBinTrans, to reflect the system's reliability.
The reviewer finds the improvements promising and the results suggest that the proposed system preserves code semantics across ISAs effectively,
leading to improved malware detection capabilities.

**Weaknesses:**

1. Prior work:

- The paper lacks coverage of several related prior works. While citing research from 1990, 2000, and 2021 is commendable,
there are more recent publications in this area. A few examples are provided in the related references below,
covering both static and dynamic translation.

- The reviewer emphasizes that ML-based code translation is an emerging but not entirely
new area of research. The comparison between the current
work and previous studies (e.g., works cited as [7], [9], [11], [12]) is
essential to position this research within the broader cybersecurity field.
It is crucial not to ignore or overlook prior work published in top security conferences.


2. UnsuperBinTrans:

The paper mentions that ``UnsuperBinTrans" is the first and only existing work. This is actually related to the prior comments.
However, without comparing or contrasting this system with other relevant studies, it is difficult to justify such a statement.

3. BPE merge times:

- How practical are the two principles utilized for BPE merge times?
- Would vocabulary size discrepancies not persist in practice? Or is the reviewer missing something?
- The paper recommends that each ISA’s vocabulary size be <12K. Is there any supporting evidence for this recommendation?

4. Model details, Training, and Testing

- The reviewer suggests including more details about the model architecture and hyperparameters used.
- What are the final evaluation losses for training across different ISAs?
- How many parameters does the model have, given that training takes approximately one day?
- Why was 4-gram precision selected? Why not explore higher precision levels?


5. Translation quality

- The translation quality seems suboptimal, with the highest BLEU score being 0.40.
- The reviewer feels the paper relies too heavily on BLEU as the sole metric.
- A comparison with the translation quality reported in other works, such as [11] and [12], would provide valuable insights.

6. Malware detection

- The model description lacks sufficient detail. The reviewer recommends including more information on the model, data preprocessing, hyperparameters, and other technical aspects.
- An error range for the reported results would enhance the clarity and reliability of the findings.

7. Discussion

- The reviewer suggests to discuss the scalability issue of the proposed system and how the system can be
 adapted to handle the packed or obfuscated malware.



Related references:

A. Static and Dynamic Translation

[1] Di Federico, Alessandro, and Giovanni Agosta. "A jump-target identification method for multi-architecture static binary translation." Proceedings of the International Conference on Compilers, Architectures and Synthesis for Embedded Systems. 2016.

[2] Gouicem, Redha, et al. "Risotto: a dynamic binary translator for weak memory model architectures." Proceedings of the 28th ACM International Conference on Architectural Support for Programming Languages and Operating Systems, Volume 1. 2022.

[3] You, Yi-Ping, Tsung-Chun Lin, and Wuu Yang. "Translating AArch64 floating-point instruction set to the x86-64 platform." Workshop Proceedings of the 48th International Conference on Parallel Processing. 2019.

[4] Chen, Jiunn-Yeu, et al. "On static binary translation of arm/thumb mixed isa binaries." ACM Transactions on Embedded Computing Systems (TECS) 16.3 (2017): 1-25.

[5] Jiang, Jinhu, et al. "A System-Level Dynamic Binary Translator Using Automatically-Learned Translation Rules." 2024 IEEE/ACM International Symposium on Code Generation and Optimization (CGO). IEEE, 2024.

[6] Xie, Benyi, et al. "An Instruction Inflation Analyzing Framework for Dynamic Binary Translators." ACM Transactions on Architecture and Code Optimization 21.2 (2024): 1-25.


B. ML and Code Translation

[7] Wang, Junzhe, et al. "Can a Deep Learning Model for One Architecture Be Used for Others? Retargeted-Architecture Binary Code Analysis." USENIX Security 2023.

[8] Ma, Xiaoyue, Lannan Luo, and Qiang Zeng. "From One Thousand Pages of Specification to Unveiling Hidden Bugs: Large Language Model Assisted Fuzzing of Matter IoT Devices." USENIX Security 2024.

[9] Wang, Junzhe, Qiang Zeng, and Lannan Luo. "Learning Cross-Architecture Instruction Embeddings for Binary Code Analysis in Low-Resource Architectures." NAACL 2024.

[10] Xie, Danning, et al. "ReSym: Harnessing LLMs to Recover Variable and Data Structure Symbols from Stripped Binaries." CCS 2024.

[11] Zuo, Fei, et al. "Neural Machine Translation Inspired Binary Code Similarity Comparison beyond Function Pairs." NDSS 2019.

[12] Roziere, Baptiste, et al. "Unsupervised translation of programming languages." Advances in neural information processing systems 33 (2020): 20601-20611.

C. Malware detection

[13] Anderson, Hyrum S., and Phil Roth. "EMBER: an open dataset for training static pe malware machine learning models." arXiv preprint arXiv:1804.04637 (2018).

**Questions:**

1. How does this work compare and contrast with the related works mentioned ([7], [9], [11], [12])?
2. How can the paper justify the statement that UnsuperBinTrans is the first and only existing work without comparing or contrasting with other relevant works mentioned in the relevant references?
3. How practical are the two principles utilized for BPE merge times?
4. In reality, would a vocabulary size discrepancy exist unless the reviewer is missing something?
5. Is there supporting evidence to back the recommendation that "The vocabulary size of each ISA is recommended to be <12K"?
6. What are the final evaluation losses for the training of each ISA?
7. How many parameters does the model have if the training takes around one day?
8. Why did the paper utilize 4-gram precision? Why not higher than 4-gram precision?
9. How does the translation quality compare with other works such as [11] and [12]?
10. Should the reported results include an error range?

---

> ### Author Response · Authors · 2024-11-17
> **Response to Reviewer yy4t**
>
> Thank you for the insightful and valuable comments. Please find our response below.
>
> > 1. *How does this work compare and contrast with the related works mentioned ([7], [9], [11], [12])?*
>
> **Response:** The works in [7] and [9] share the same goal as ours: enabling the reuse of a model trained on a high-resource ISA for low-resource ISAs to address data scarcity. However, [7] and [9] achieve this by learning cross-architecture instruction embeddings, whereas our approach focuses on translating binary code across ISAs. By translating code to a high-resource ISA, our method allows direct application of existing downstream models—trained on the high-resource ISA—to other ISAs through testing the translated code. In contrast, [7] and [9] require retraining the model using the cross-architecture instruction embeddings. Additionally, translating code from one ISA to another helps human analysts understand code from unfamiliar ISAs, supporting broader code comprehension applications.
>
> The work in [11] applies neural machine translation techniques for binary code similarity comparison but does not perform binary code translation across ISAs. It uses only the encoder of a neural machine translation model to generate embeddings for binary code pairs, and measures similarity based on embedding distance. In contrast, our work translates binary code across ISAs.
>
> The work in [12] addresses source code translation (e.g., C to Java), rather than binary code translation (e.g., assembly code from x86-64 to ARM32). Thus, it targets different languages and serves different applications than our approach.
>
> > 2. *How can the paper justify the statement that UnsuperBinTrans is the first and only existing work without comparing or contrasting with other relevant works mentioned in the relevant references?*
>
> **Response:** We apologize for the confusion. Our intent was to state that UnsuperBinTrans is the first and only existing work focused on binary code translation by leveraging deep learning techniques. While we acknowledge other approaches for binary code translation, as discussed in Section 2.3, these rely on program analysis techniques rather than deep learning. Similarly, as noted in Section 2.2 and [12], there are works on source code translation, but these are not designed for binary code. Thus, UnsuperBinTrans remains the only existing work that applies deep learning techniques to binary code translation. We will revise the paper to make it clear.
>
> > 3. *How practical are the two principles utilized for BPE merge times?*
>
> **Response:** The two principles are derived from both empirical experiments and theoretical insights discussed in prior research.
>
> The first principle (vocabulary size discrepancy < 15% between source and target ISAs) ensures relative class balance in neural machine translation. Prior research [14] highlights that class imbalance can degrade translation quality. In neural machine translation, large vocabulary size discrepancies can create an imbalanced learning problem, where the model may allocate disproportionate attention to the larger vocabulary, potentially leading to inefficiencies, overfitting, or difficulties in generalizing to less-represented vocabularies.
>
> The second principle (vocabulary size < 12k per ISA) strikes a balance between semantic representation and computational efficiency. Prior research [15] demonstrates that larger vocabulary sizes can negatively affect model performance due to increased complexity and sparse token distributions.
> We acknowledge that these principles are tailored to our specific scenarios.
>
> > 4. *In reality, would a vocabulary size discrepancy exist unless the reviewer is missing something?*
>
> **Response:** Yes, a vocabulary size discrepancy (> 15%) may exist. If the vocabulary size of one ISA is significantly larger (or smaller) than that of another, a subset of the vocabulary from this ISA (or the other) may be selected for training MALTRANS to address the imbalance.
>
> > 5. *Is there supporting evidence to back the recommendation that "The vocabulary size of each ISA is recommended to be <12K"?*
>
> **Response:** This recommendation is based on empirical experiments. We tested different preprocessing rules, each influencing the vocabulary size, and observed that limiting the vocabulary size of each ISA to less than 12K produced the best translation performance. Additionally, prior research [15] shows that larger vocabulary sizes can negatively impact model performance due to increased complexity and sparse token distributions. The threshold of 12K was determined empirically as the optimal balance between model performance and computational efficiency.
>
> > 6. *What are the final evaluation losses for the training of each ISA?*
>
> **Response:** For each ISA pair, we trained the translation model until the evaluation loss dropped below 0.5.
>
> Please find our continued response in the following official comment.

---

> > ### Author Response · Authors · 2024-11-17
> > **Continued Response to Reviewer yy4t**
> >
> > > 7. *How many parameters does the model have if the training takes around one day?*
> >
> > **Response:** The model comprises the following number of parameters: the encoder has 788,190 parameters, while the source decoder and the target decoder have 855,262 parameters each.
> >
> > Training time is also influenced by the size of the training datasets. For each ISA, the mono-architecture datasets contain the following number of basic blocks: 2,789,119 for X86-64, 2,803,557 for i386, 7,413,083 for ARM64, 5,812,795 for ARM32, and 5,365,474 for s390x.
> >
> > > 8. *Why did the paper utilize 4-gram precision? Why not higher than 4-gram precision?*
> >
> > **Response:**  Apologies for the confusion. We do not use 4-gram precision exclusively. Instead, we calculate the BLEU score as the average of unigram, bigram, trigram, and 4-gram precision.
> >
> > We did not consider n-grams higher than 4-grams for the following reasons. (1) Higher n-grams (5+ grams) become increasingly sparse. This issue is exacerbated in code translation, where instruction sequences are typically shorter than natural language sentences, making higher n-grams less reliable as evaluation metrics. (2) The original BLEU metric proposed by [16] demonstrated that 4-gram BLEU provides an optimal balance between accuracy and computational efficiency. Beyond 4-grams, the benefits diminish due to increased computational costs and reduced reliability.
> >
> > > 9. *How does the translation quality compare with other works such as [11] and [12]?*
> >
> > **Response:** The work in [11] utilizes neural machine translation techniques for binary code similarity comparison but does not perform binary code translation across ISAs. Specifically, it employs only the encoder of a neural machine translation model to generate embeddings for two pieces of binary code, and measures similarity based on embedding distance. In contrast, our work focuses on translating binary code across different ISAs.
> >
> > The work in [12] focuses on source code translation (e.g., C to Java), which differs significantly from binary code translation (e.g., assembly code from x86-64 to ARM32).
> >
> > As the objectives and languages differ, a direct comparison of translation quality is not applicable.
> >
> > > 10. *Should the reported results include an error range?*
> >
> > **Response:** We will include the error range in the revision.
> >
> > > 11. *discuss the scalability issue of the proposed system and how the system can be adapted to handle the packed or obfuscated malware.*
> >
> > **Response:** The training time for the four pairs of ISAs is approximately 23h, 22h, 25h, and 22h, respectively. However, this training is a one-time effort. The average time for translating a basic block from one ISA to x86-64 is 0.01s. Thus, our approach demonstrates good scalability.
> >
> > To address packed malware, we will incorporate advanced unpacking tools, such as PEiD and OllyDbg, to first unpack the malware and then analyze the unpacked content. For obfuscated malware, one potential solution is to train MALTRANS on obfuscated samples, which can be generated using existing obfuscation tools like Trigress and LLVM-obfuscator, enabling MALTRANS to learn how to handle obfuscated code. Alternatively, we can apply deobfuscation techniques, which dynamically analyze the execution flow by using both static and behavioral analysis to identify obfuscated code and reconstruct the original payload. We will include a discussion in the revision.
> >
> >
> > In the revision, we will include more details about  the model architecture, data preprocessing, and hyperparameters.
> >
> >
> >  [14] Thamme Gowda and Jonathan May. 2020. Finding the Optimal Vocabulary Size for Neural Machine Translation. In Findings of the Association for Computational Linguistics: EMNLP 2020, pages 3955–3964, Online. Association for Computational Linguistics.
> >
> > [15] Jean, S., Cho, K., Memisevic, R., & Bengio, Y. (2014). On Using Very Large Target Vocabulary for Neural Machine Translation. Proceedings of the 53rd Annual Meeting of the Association for Computational Linguistics and the 7th International Joint Conference on Natural Language Processing.
> >
> > [16] Kishore Papineni, Salim Roukos, Todd Ward, and Wei-Jing Zhu. 2002. BLEU: a method for automatic evaluation of machine translation. In Proceedings of the 40th Annual Meeting on Association for Computational Linguistics (ACL '02).

---

### Official Review · Reviewer_FiFn · 2024-11-03

**Soundness:** 3
**Presentation:** 3
**Contribution:** 2
**Rating:** 5
**Confidence:** 4

**Summary:**

This paper introduces an unsupervised binary code translation model (MalTrans) designed for malware detection across multiple ISAs. The main challenge tackled in this work is the unavailability of large data samples from ISAs such as i386, ARM64, ARM32, and s390x. This approach also operates without needing labeled malware samples for code translation. Experimental results show that after code translations, for an LSTM malware detection model, this archives AUC scores of 0.999, 0.986, 0.913, and 0.938 for i386, ARM32, ARM64, and s390x, respectively. This outperforms the two baselines (UnsuperBinTrans, IR-based Model).

**Strengths:**

1. Novelty:

a. Uses a transformer-based model compared to the recent work using RNNs (UnsuperBinTrans)
b. Uses 3 different instruction normalization techniques (R1, R2, R3)
c. Experiments were conducted using multiple ISAs compared to previous work

2. The proposed approach achieves relatively better  BLEU scores and retains high AUC values in malware detection across different ISAs.

3. The main outcome of this work is the utilization of knowledge transfer across ISAs for malware detection: Doing code translation from ISA-2 to ISA-2. Have a trained model on ISA-2 with more data. Test for ISA-2 using the translated code for malware detection.

4. The paper is well written with minimal mistakes. The structure is good.

**Weaknesses:**

1. Malware detection is a downstream task that the paper uses to show the evaluations. The paper does not do any innovation concerning malware detection. Similar to malware detection, any other task can be used here. The main innovation here seems to be a better translation mechanism of code compared to UnsuperBinTrans.

2. Among the contributions claimed in the paper, similar contributions were made in UnsuperBinTrans. The novelty is the model architecture, normalization scheme, and availability for more ISAs.

3. Assessment for s390x does not seem to be fair since there are only 2 malware samples (line 410).

4. It seems that the testing data sets are not consistent between Table 3 (a) and (b) for ARM65 and s390x. So the accuracies obtained cannot be directly compared.

**Questions:**

1. In Table 3 (b), it seems there results are for LSTM models with non-translated data. For example, according to the results, it is possible to get up to 0.999 with as little as 265 data samples for i386. 0.988 for ARM32 with 196 data samples. Does this contradict the initial claim of the paper? Do we actually need to go through a complex code translation process when we can get good results with 265 data points?

2. It would be interesting to see BLEU scores for MALTRANS models with smaller datasets. What would be the impact on the accuracy if few-shot learning is done? What should be the ideal ratio of data for 2 ISAs? Also, can we get better BELU scores if we have more data samples? Are these BELU scores for 1-gram?

3. It would be better if it is possible to provide numbers for UnsuperBinTrans for other ISAs although it only focused on ARM32→X86. It should be trivial and a good comparison to do. Any possibility?

4. Why the accuracy results are better for ARM32→X86 although the BELU score is higher for ARM64→X86? Any explanation for that?

5. It would be ideal to compare the performance of malware detection using classical machine learning techniques as a comparison with LSTM. Can that be done?

---

> ### Author Response · Authors · 2024-11-17
> **Response to Reviewer FiFn**
>
> Thank you for the insightful and valuable comments. Please find our response below. We are currently conducting evaluations related to items 3, 6 and 8. We plan to provide updated results by November 26. The final revision will include all updated findings.
>
> > 1. *Malware detection is a downstream task that the paper uses to show the evaluations. The paper does not do any innovation concerning malware detection. Similar to malware detection, any other task can be used here. The main innovation here seems to be a better translation mechanism of code compared to UnsuperBinTrans.*
>
> > *Among the contributions claimed in the paper, similar contributions were made in UnsuperBinTrans. The novelty is the model architecture, normalization scheme, and availability for more ISAs.*
>
> **Response:** Our contribution focuses on advancing code translation across ISAs, enabling malware detection on ISAs with limited labeled malware samples. We do not claim that MALTRANS introduces a new malware detection model; rather, its main contribution is on code translation to support malware detection, addressing the data scarcity challenge that would otherwise hinder robust detection for low-resource ISAs.
>
> Compared to UnsuperBinTrans, MALTRANS achieves better malware detection across ISAs due to its superior translation capability, enhanced by the model architecture and normalization schemes we developed. Furthermore, we expanded the evaluation to cover more ISAs, showcasing MALTRANS’s broader applicability and improved performance.
>
> > 2. *Assessment for s390x does not seem to be fair since there are only 2 malware samples.*
>
> **Response:** We collected all available malware samples from VirusShare.com, a widely used malware repository, which provided only 2 s390x samples (after deduplication). This limitation actually underscores the significance of our work: directly training a robust malware detection model on only 2 s390x samples is impractical, but MALTRANS allows us to translate s390x binaries to x86-64, enabling the use of an x86-64-trained model to analyze the translated code for malware detection.
>
> We will clarify in the revision that the s390x results primarily illustrate MALTRANS's adaptability in low-resource conditions, and that future work will aim to include a larger s390x dataset for a more comprehensive evaluation.
>
> > 3. *It seems that the testing data sets are not consistent between Table 3 (a) and (b) for ARM64 and s390x. So the accuracies obtained cannot be directly compared.*
>
> **Response:** Thank you for noting this. You are correct. In Table 3(a), we use all available malware samples in each of the four ISAs for testing, while in Table 3(b), we use 20% of the malware samples for testing (reserving 80% for training the optimal model). We will reconduct the experiments and provide the results, ensuring that only 20% of malware samples across the four ISAs are translated by MALTRANS and tested by the x86-64-trained LSTM model for consistency with Table 3(b).
>
>  > 4. *Do we actually need to go through a complex code translation process when we can get good results with 265 data points?*
>
> **Response:**  There are two kinds of ISAs: high-resource ISAs and low resource ones. Our work does not aim at high-resource ISAs. Instead, we aim to address scenarios where malware samples are insufficient for certain ISAs like ARM64 and s390x, necessitating translation to resolve the data scarcity issue. For example, for the low-resource ISA s390x, after collecting all available malware samples from VirusShare.com, we were only able to obtain 2 s390x samples. We will clarify this in the revision.
>
>  > 5. *What would be the impact on the accuracy if few-shot learning is done? What should be the ideal ratio of data for 2 ISAs? Also, can we get better BELU scores if we have more data samples? Are these BELU scores for 1-gram?*
>
> **Response:** Few-shot learning could potentially improve accuracy. However, since our focus is on unsupervised learning, we did not use labeled code samples for training MALTRANS. We leave it as future work.
>
> In general, increasing the amount of high-quality data samples can lead to improved BLEU scores. However, the relationship between data size and BLEU score is not always linear. Vocabulary coverage plays an important role. We assessed the adequacy of our mono-architecture datasets by studying vocabulary growth as we incrementally include more programs. Our findings indicate that while the vocabulary size increases with additional programs, it stabilizes after a certain point. Based on this vocabulary growth trend, our datasets are sufficient to cover the necessary instructions and support effective code translation.
>
> The current BLEU scores reflect an average of unigram, bigram, trigram, and 4-gram precision. We will clarify this in the revision.
>
> Please find our continued response in the following official comment.

---

> > ### Author Response · Authors · 2024-11-17
> > **Continued Response to Reviewer FiFn**
> >
> > > 6. *It would be better if it is possible to provide numbers for UnsuperBinTrans for other ISAs although it only focused on ARM32→X86. It should be trivial and a good comparison to do. Any possibility?*
> >
> > **Response:** We will attempt to train UnsuperBinTrans on our datasets for additional ISA pairs and include the results in the revision.
> >
> > >7. *Why the accuracy results are better for ARM32→X86 although the BELU score is higher for ARM64→X86? Any explanation for that?*
> >
> > **Response:** While the BLEU score and downstream task performance are related, they do not always show a strictly positive correlation. The BLEU score measures n-gram overlap between translated and reference code, indicating translation quality based on structural similarity. However, downstream malware detection depends more on preserving code semantics rather than exact n-gram matches. This may explain why ARM32→X86 achieves higher detection accuracy despite a lower BLEU score, as the translated ARM32 code may better retain semantic features relevant to detection. We will clarify this in the revision.
> >
> > >8. *It would be ideal to compare the performance of malware detection using classical machine learning techniques as a comparison with LSTM. Can that be done?*
> >
> > **Response:**  Thanks for the suggestion. We will explore other models such as CNNs or Transformers, and will include these results in the revised paper.

---

> > > ### Comment · Reviewer_FiFn · 2024-11-25
> > >
> > > Thank you for your detailed response and clarifications. I appreciate your effort to address my concerns. I belive the additional experiments will add more depth to the paper making it stronger. However, my concerns about limited novelty compared to UnsuperBinTrans remain, and UnsuperBinTrans already achieved some of the claimed contributions. Therefore, I will maintain my original scores.

---

> > > > ### Author Response · Authors · 2024-11-25
> > > > **Response to Official Comment by Reviewer FiFn**
> > > >
> > > > Thank you for your kind response, and we apologize for not clarifying this earlier. Below, we summarize the key novelties of MalTrans:
> > > >
> > > > * **Enhanced Normalization Rules:** MalTrans introduces new normalization rules for assembly code, which differ significantly from those used by UnsuperBinTrans. Specifically, normalization rules R1 and R2, which address issues with dummy names generated by IDA Pro and normalize function names, are not applied by UnsuperBinTrans. The absence of these rules results in numerous out-of-vocabulary words during testing, potentially degrading translation performance. Table 4 illustrates the significant impact of R1 and R2 on malware detection performance.
> > > >
> > > > * **Application to Malware Detection:** UnsuperBinTrans has not been previously applied to the malware detection task, leaving its effectiveness in this domain uncertain. As shown in Table 3(a), when UnsuperBinTrans is applied to malware detection, it performs poorly. This could be due to its unsuitable normalization rules and less effective model architecture. In contrast, MalTrans achieves superior malware detection across ISAs, thanks to its improved model architecture and tailored normalization schemes.
> > > >
> > > > * **Broader ISA Coverage:** While UnsuperBinTrans is limited to two ISAs (x86-64 and ARM 32), MalTrans extends the evaluation to a wider range of ISAs, demonstrating both broader applicability and improved performance.
> > > >
> > > > We hope this clarifies the unique contributions of MalTrans compared to UnsuperBinTrans. We will clarify this in the revision.

---

### Official Review · Reviewer_DKbR · 2024-11-04

**Soundness:** 2
**Presentation:** 2
**Contribution:** 3
**Rating:** 5
**Confidence:** 3

**Summary:**

This work proposes an architecture that utilizes an unsupervised binary code translation model called
MALTRANS to translate binaries from four different Instruction Set Architectures (ISAs) into a target ISA
(X86-64). A deep learning model trained exclusively on the target ISA is then employed for malware
detection.

**Strengths:**

**Highly relevant topic.** With the growing number of devices targeted by malicious software (malware),
especially within IoT devices, which come in a wide variety of architectures, the need for cross-ISA malware
detection systems is increasingly critical. This study addresses this need by proposing a detection system
capable of handling multiple ISAs.

**Novelty.** The subject is relatively novel, with the current state of the art lacking extensive work on cross-ISA
translation. Additionally, the proposed work could significantly contribute to the development of efficient
cross-platform malware detection systems.

**Good number of analyzed ISAs.** This study covers a good number of ISAs relevant to todayʼs market,
particularly focusing on Linux environments and IoT devices.

**Weaknesses:**

**Small malware dataset.** The malware detection dataset used by the authors is quite limited. Specifically,
for the X86-64 architecture analysis, only 1,238 malware samples were used, divided into an 80/20 traintest
split, which is insufficient given the prevalence of this architecture. Additionally, while the focus is on
the Linux and IoT landscape, the ARM architecture datasets were also notably small, despite the broader
sample availability in the literature. Given the importance of the topic, it would be recommended to conduct
experiments on a more extensive dataset to ensure more reliable results.

**Unclear description.** MALTRANS is introduced as an ISA-to-ISA translation tool and presented as such in
Figure 1, but later, it is also described as capable of detection. Clarification is needed to delineate its
functions. Does MALTRANS perform solely translation or detection as well? Better differentiation between
the two modules would enhance clarity.
Figure 2 shows the MALTRANS architecture; however, the flow is not entirely clear. For example, the text
mentions inputs like B'_src, but in the figure, B_src is shown instead of its noisy version. During stages (3
and 4), outputs B_tgt^ and B_src^ are produced, but the workflow suggests they are not subsequently
used, which conflicts with the description stating these outputs are fed into the model to predict the original
block.
In the paragraph "Comparison with Optimal Model," the dataset usage is unclear. Training and testing on as
few as 58 or even 2 samples divided 80/20 is suboptimal. Providing a clear dataset usage methodology
before discussing the results of the optimal model would help.


**Experiments performed on a single deep learning model.** While the chosen model is acceptable, there
are several state-of-the-art malware detection models for X86-64 that could be considered. The use of
pre-trained models in the detection module could yield a fairer comparison.
The acronyms for BLEU, AUC, and F1-score are not expanded. Although they are widely recognized, it
would be helpful to describe each acronym and provide brief descriptions.

**Typographical error (minor issue).** In section 4.2, the title "MODE ARCHITECTURE" likely should be "MODEL
ARCHITECTURE."

**Questions:**

Please comment on the main weaknesses:

- Experiments performed using a single deep learning model;
- Absence of the MIPS ISA;
- Limited malware dataset;
- Unclear methodological description.

---

> ### Author Response · Authors · 2024-11-17
> **Response to Reviewer DKbR**
>
> Thank you for the insightful and valuable comments. Please find our response below. We are currently conducting evaluations related to items 1, 3, and 4. We plan to provide updated results by November 26. The final revision will include all updated findings.
>
> > 1. *Small malware dataset.*
>
> **Response:** Thank you for pointing this out. We are currently expanding the malware datasets and will conduct experiments with these larger datasets.
>
> > 2. *Unclear description.*
>
> **Response:** We apologize for the confusion. MALTRANS is designed for ISA-to-ISA translation. The malware detection capability is provided by the LSTM model, which analyzes assembly code translated by MALTRANS to detect malware. We will revise the paper to clarify this point.
>
> Regarding Figure 2, thank you for pointing out the potential confusion. We will revise this section to include two separate figures illustrating how denoising auto-encoding and back translation are employed to train MALTRANS, which should clarify the workflow.
>
> In the "Comparison with Optimal Model" paragraph, for x86-64, i386, and ARM32, we used an 80/20 split for training and testing. For ARM64 and s390x (this is not as popular as x86-64 and ARM32), however, we could only obtain 58 and 2 malware samples, respectively, from VirusShare.com (after deduplication). We used 46 ARM64 and 1 s390x samples for training and 12 ARM64 and 1 s390x samples for testing. While these sample sizes are limited, they reflect the value of MALTRANS: by translating code from a low-resource ISA to a high-resource ISA, we can leverage models trained on the high-resource ISA, addressing data scarcity challenges that would otherwise hinder robust detection for low-resource ISAs.
>
> > 3. *Experiments performed on a single deep learning model.*
>
> **Response:** Thank you for raising this excellent point. The primary novelty and main contribution of this work lie in translating code from a low-resource ISA to a high-resource ISA. This approach enables even a relatively simple model, such as LSTM, to achieve strong performance.
>
> We will explore other models such as CNNs or Transformers, and will include these results in the revised paper.
>
> > 4. *Absence of the MIPS ISA.*
>
> **Response:** Thank you for this suggestion. We are currently working on adding MIPS to our evaluation and are in the process of collecting MIPS malware samples, which will be included in the revision.
>
> Thank you for pointing out the typo. We will carefully revise the paper and include descriptions of BLEU, AUC, and F1-score.

---

> > ### Comment · Reviewer_DKbR · 2024-12-02
> > **Ack**
> >
> > Thank you for your response. I see that the authors are working to improve their paper, but at the moment, the major weaknesses remain unaddressed, so I'll keep my score.

---

> > > ### Author Response · Authors · 2024-12-02
> > > **Response to Ack**
> > >
> > > Thank you for your response. Could you kindly specify which weaknesses remain unaddressed? We believe we have addressed all the concerns raised in your review, as detailed in the "**Summary of Changes**" file included at the beginning of the submitted revision.
> > >
> > > For your convenience, we have also summarized the changes in the comments below.
> > >
> > > All the changes made in response to Reviewer DKbR’s comments are highlighted in **Blue**.
> > >
> > > > 1. Reviewer's Comment: “*Small malware datasets.*”
> > >
> > > **Authors’ Response:**
> > > We have expanded the malware datasets and conducted experiments using these larger datasets. Specifically, we increased the malware samples for the X86-64, i386, ARM32, ARM64, MIPS32, and s390x from 1238, 265, 196, 58 and 2 to 2140, 1740,1581, 58, and 2, respectively. Note that for ARM64 and s390x (this is not as popular as x86-64 and ARM32), however, we connected all malware samples from VirusShare.com, but could only obtain 58 and 2 malware samples, respectively (after deduplication). All experimental results have been updated accordingly, as detailed in **the Evaluation Section**.
> > >
> > > ----------------------------------------------------------------------------------------------------------------------------
> > >
> > > > 2. Reviewer's Comment:
> > > “*Unclear description. MALTRANS is introduced as an ISA-to-ISA translation tool and presented as such in Figure 1, but later, it is also described as capable of detection. Clarification is needed to delineate its functions.” “Figure 2 shows the MALTRANS architecture; however, the flow is not entirely clear.” “In the paragraph "Comparison with Optimal Model," the dataset usage is unclear.*”
> > >
> > > **Authors’ Response:**
> > > MALTRANS is designed for ISA-to-ISA translation. The malware detection capability is provided by the LSTM model, which analyzes assembly code translated by MALTRANS to detect malware. We have revised **the caption of Table 3 (Line 378)** to clarify this point.
> > >
> > > We have replaced the original **Figure 2** with two separate subfigures (**Line 216 to Line 223**) that illustrate how denoising auto-encoding and back translation are employed to train MALTRANS. The presentation in **Section 4.4** has also been revised accordingly to align with these changes.
> > >
> > > In the "Comparison with Optimal Model" paragraph, for x86-64, i386, and ARM32, we used an 80/20 split for training and testing. For ARM64 and s390x (this is not as popular as x86-64 and ARM32), however, we could only obtain 58 and 2 malware samples, respectively, from VirusShare.com (after deduplication). We used 46 ARM64 and 1 s390x samples for training and 12 ARM64 and 1 s390x samples for testing. While these sample sizes are limited, they reflect the value of MALTRANS: by translating code from a low-resource ISA to a high-resource ISA, we can leverage models trained on the high-resource ISA, addressing data scarcity challenges that would otherwise hinder robust detection for low-resource ISAs. We have revised the “**Task-Specific Training Dataset” and “Task-Specific Testing Dataset” parts in Section 5.4 (Line 401 to Line 422)** to clarify this.
> > >
> > > ----------------------------------------------------------------------------------------------------------------------------
> > >
> > > >3. Reviewer's Comment:
> > > “*Experiments performed on a single deep learning model.*”
> > >
> > > **Authors’ Response:**
> > > We have utilized CNNs for malware detection and included the detection results in **Appendix E (Line 803)**.
> > >
> > > ----------------------------------------------------------------------------------------------------------------------------
> > >
> > > >4. Reviewer's Comment:
> > > “*Absence of the MIPS ISA.*”
> > >
> > > **Authors’ Response:**
> > > We have collected 1430 MIPS malware samples and included them in the evaluation. The revised results include updates to the vocabulary size (**Table 1; Line 424**), BLEU scores (**Table 2; Line 332**), malware detection results (**Table 3; Line 378**), and the hyperparameter study (**Tables 4 and 5; Line 486 and Line 494**). The presentation has been revised accordingly to reflect these updates.
> > >
> > > ----------------------------------------------------------------------------------------------------------------------------
> > >
> > > > 5. Reviewer's Comment:
> > > “The acronyms for BLEU, AUC, and F1-score are not expanded. Although they are widely recognized, it would be helpful to describe each acronym and provide brief descriptions.”
> > >
> > > **Authors’ Response:**
> > > We have included the descriptions of BLEU score in **Line 355** and AUC/F1-score in **Line 420**.
> > >
> > > Thanks for your time!

---

> > > > ### Comment · Reviewer_DKbR · 2024-12-02
> > > > **Reply**
> > > >
> > > > I appreciate your efforts in improving the work, but I remain unconvinced regarding the experimental results with one architecture and (still) small datasets. I also see that the other reviews align with my evaluation, so for now, I'll keep my score.

---

> ### Author Response · Authors · 2024-12-02
> **Response**
>
> Thank you for your response. We plan to open-source the code, dataset, and trained model once the paper is accepted.
>
> Moreover, the size of our dataset is comparable to that of UniMap (USENIX Security '23). UniMap utilizes 2,156, 1,159, and 1,004 malware samples for x86-64, ARM32, and MIPS32, respectively, while our dataset includes 2,140, 1,740, and 1,581 samples for these architectures.
>
> Additionally, we expanded our dataset to include two more ISAs: ARM64 and s390x. However, the two ISAs are less popular than x86-64, and despite collecting all available malware samples from VirusShare.com, we could only obtain 58 and 2 samples, respectively.

---

### Author Response · Authors · 2024-11-26
**Response to All Reviewers**

We would like to thank the reviewers for the insightful comments and constructive suggestions. We have uploaded the **revised paper**, along with a file titled "**Summary of Changes**," which is placed before the revised paper.

We are delighted that **ALL** reviewers recognize the **practical significance** of our *cross-ISA binary code translation* work, which makes an important contribution to detecting malware across ISAs by leveraging a model trained on a high-resource ISA (X86-64), effectively addressing the data scarcity challenge of low-resource ISAs. We are also encouraged by the reviewers’ acknowledgment of the **novelty** of this work and its ability to fill gaps in the current state of the art (Reviewers DKbR and FiFn), its coverage of **multiple ISAs** (Reviewers DKbR, FiFn, and yy4t), the value of **unsupervised learning** (Reviewer yy4t), the **improved performance** (Reviewers FiFn and yy4t), and the **clear interpretation and explanation** of how and why our model evolves (Reviewer C5ox). Additionally, we appreciate the positive feedback on the quality of writing (Reviewer FiFn).

We first summarize the newly added experiments as follows.

* Expanded the malware datasets and conducted all experiments using these larger datasets (Reviewer DKbR).
* Utilized CNNs for malware detection and presented the detection results (Reviewer DKbR and FiFn).
* Included the MIPS ISA in the evaluation (Reviewer DKbR).
* Trained the baseline model, UnsuperBinTrans, on the additional ISAs, and compared its performance with our model (Reviewer FiFn).

In addition to the new experiments, we have carefully revised the submission to **address all reviewers’ comments and concerns**. Detailed responses and changes for each reviewer comment are provided in the "**Summary of Changes**" file.

Please let us know if you have any further questions or comments. Thank you!

---

### Meta-Review · Area_Chair_UUp4 · 2024-12-15

**Metareview:**

Summary:
This paper addresses the challenge of cross-ISA malware detection by proposing neural machine translation to translate malware between ISAs, reducing the need for large datasets. The approach aims to enable efficient cross-platform malware detection through data augmentation.

Strengths:
The paper tackles a novel and impactful problem, proposing an innovative use of translation for data augmentation, which could significantly improve cross-platform malware detection systems.

Weaknesses:
The evaluation dataset is small, with only 58 and 2 samples for ARM64 and s390x, respectively. This undermines the robustness and generalizability of the approach. Practical scalability and comparisons with other methods are also lacking. Since using translation for data augmentation has been studied in other domain, there is limited novelty on the algorithm itself, the key contribution of this paper would be on malware application, yet the data is not sufficient. The model is only one architecture, limiting the result of the paper.

Decision:
Reject. Despite the novelty, the limited dataset and insufficient evaluation weaken the paper’s contributions and impact. A more comprehensive study is needed.

**Additional Comments On Reviewer Discussion:**

After the rebuttal, 3 of the reviewers remains negative, where for example Reviewer DKbR asked the question on the sample size and this has been left unaddressed.

---

### Decision · Program_Chairs · 2025-01-22

Reject